# Kernel von Mises Formula of the Influence Function

**Yaroslav Mukhin**
Cornell University
kif@ymx.io

## Abstract

The influence function (IF) of a statistical functional is the Riesz representer of its derivative, also known as its first variation and Fisher-Rao gradient. It is a key object for numerical optimization over probability measures, semiparametric efficiency theory, standard constructions of efficient estimators, and an arsenal of inference methods for these estimators. Yet, deriving the IF analytically is often an obstruction for practitioners. To automate this task, we develop a novel spectral representation of the IF that lends itself to a low-rank functional estimator in a reproducing kernel Hilbert space (rkHs). Our estimator (i) does not require analytic derivations by the user, (ii) relies on kernel Principal Component Analysis and numerical pathwise derivatives along these components. We present the derivation of the representation and prove consistency of the low-rank rkHs estimator.

## 1 Introduction

The target $\theta$ of a statistical learning procedure often takes the form of a mapping $P \mapsto \theta(P)$ of the probability measure $P$ on a sample space $\mathcal{X} \subset \mathbb{R}^d$ governing the training data $X_{1:n}$. The difficulty of learning $\theta$ from data depends on the combination of structural properties of all possible $P$ and the *local variability* of $\theta$ at the unknown true $P$. For example, estimating the mean $\theta_1(P) := \int x \, dP$ is easy even if $P$ is unrestricted beyond having finite moments. By contrast, estimating the density $f(x_0)$ of $P$ at a point $x_0$ is easy in a model for $P$ that is smoothly parametrized by a subset of $\mathbb{R}^p$, but much harder in nonparametric settings. The rate of convergence of an estimator $\hat{\theta}_n(X_{1:n})$ to the target $\theta(P)$ as the sample size $n$ increases is a measure of the statistical difficulty of estimating $\theta$. The rate of estimating $\theta_1$ with the sample mean is $\sqrt{n}$ by the Central Limit Theorem (CLT), whereas the rate of estimating $f(x_0)$ can range from $\sqrt{n}$ to arbitrarily slow, depending on the regularity of $P$. If $f(x)$ is smooth in a neighborhood of $x_0$ for all $P$, then observations close to $x_0$ can be aggregated, as for estimating $\theta_1$, resulting in a fast rate; if $f(x)$ varies roughly near $x_0$ and as $P$ is perturbed, then aggregation leads to bias and must be limited at the expense of precision.

We consider scalar functionals $\theta(P)$ that can be estimated at the parametric $\sqrt{n}$ rate in nonparametric models for $P$, like the mean $\theta_1$. This includes scalar and vector estimands that depend on averages of nuisance functions, e.g., $\theta_0(P) := \int f \, dP$ and estimands of causal inference, but not $f(x_0)$. For these parameters the rate is fixed and the difficulty of estimation is characterized in terms of a lower bound on the asymptotic variance. The object that determines this bound is the *influence function (IF)* of $\theta$ at $P$, denoted $\psi_P : \mathcal{X} \to \mathbb{R}$; the bound is the norm $\int \psi_P^2 \, dP$ of this function. Setting aside the interpretation of $\psi_P(x)$ until Theorem 2.1, recall when one can expect $\theta(P)$ to be estimable at the $\sqrt{n}$ rate and to have an IF. This requires that $\theta(P)$ *varies smoothly with $P$*. Specifically, the map $P \mapsto \theta(P)$ must have a derivative $D\theta_P$ that maps perturbations to $P$ into infinitesimal changes in $\theta(P)$ [Mis47; Ste56; KL76; IH81; Vaa91]. When $D\theta_P$ exists and is a bounded linear map on $L^2(P)$, Riesz's theorem guarantees the existence of a function $\psi_P \in L^2(P)$ such that for any perturbation $\phi$ to $P$, the effect on $\theta(P)$ is given by $D\theta_P[\phi] = \int \phi \psi_P \, dP$; we review this further in Theorem 2.1. If $\theta(P)$ is not smooth in $P$, so the derivative $D\theta_P$ doesn't exist or is unbounded and has no IF, then the estimation rate for $\theta(P)$ is typically slower than $\sqrt{n}$.

39th Conference on Neural Information Processing Systems (NeurIPS 2025).

**Example 1.1.** The IF for $\theta_1(P)$ is $\psi_1(x) = x - \theta_1$; the IF for $\theta_0(P)$ is $\psi_0(x) = 2[f(x) - \theta_0]$. With $q \in (0,1)$, the $q$-quantile is $\theta_q(P) := F^{-1}(q)$, so that $F(\theta_q) := P(X \leq \theta_q) = q$, and its IF is $\psi_q(x) = [q - \mathbb{1}(x \leq \theta_q)]/f(\theta_q)$, where $\mathbb{1}_A$ is the indicator of event $A$. To simplify notation, we may write $\psi$ without the subscript $P$, but the IF always depends on $P$, as do the functionals $\theta$ and $D\theta$.

Consider now an estimator sequence $\hat{\theta}_n = \hat{\theta}_n(X_{1:n})$ for $\theta(P)$ constructed with i.i.d. samples $X_{1:n}$ from $P$. It is *regular* if $\sqrt{n}(\hat{\theta}_n - \theta(P))$ converges in distribution, and, furthermore, the limit law is invariant to vanishing perturbations in $P$; a counterexample is hard thresholding. An estimator is asymptotically linear if it has the representation analogous to the CLT:

$$\sqrt{n}(\hat{\theta}_n - \theta(P)) = \frac{1}{\sqrt{n}} \sum_{i=1}^{n} \psi_P(X_i) + o_P(1), \quad \text{where} \quad \psi_P \in L_0^2(P) \tag{1.1}$$

meaning $\int \psi_P^2 \, \mathrm{d}P < +\infty$ and $\int \psi_P \, \mathrm{d}P = 0$, and the $o_P(1)$ term vanishes in $P$ as $n \to \infty$. Note that $\psi_P$ in (1.1) is the IF of $\theta$, also called the IF of the estimator $\hat{\theta}$. [Vaa91] shows that $\theta(P)$ is differentiable in $P$ and has an IF if and only if there exists a regular asymptotically linear estimator. Moreover, [Kla87] shows that a regular asymptotically linear estimator exists if and only if the IF exists and can itself be consistently estimated. For further details of semiparametric efficiency theory we refer to [Vaa00, ch25] and [Bic+93], and now turn to a very brief and selective review of its recent methodological uses in machine learning (ML), statistics and econometrics.

According to (1.1), the contribution of a datum $X_i$ to the fluctuations in the estimator is approximately $\psi(X_i)/n$. Consequently, the observations with the most influence on the realized estimate are $X_i$ with a large $|\psi(X_i)|$ value. Returning to Example 1.1, for estimates of $\theta_1$, the influential data are those far away from the mean, i.e., the 'outliers'; by contrast, for estimates of $\theta_q$ all observations have similar influence; for estimates of $\theta_0$ the observations with extreme density values are influential. This interpretation of the IF to study robustness to outliers goes back to [Hub64] and appears recently in e.g., [Pru+20; BGM20]. In Theorem 2.1, we discuss a related interpretation of the IF, but for the functional rather than its estimators, that is an important baseline for our work.

Closely related with robustness are the uses of the IF toward data attribution and interpretability of nontransparent estimators. The idea is to approximate the $x_i \mapsto \hat{\theta}(x_{1:n})$ mapping with $x_i \mapsto \psi(x_i)$ to gain insight into the effect of a data point or statistic on the estimate; e.g., in large language [Gro+23], black box ML [KL17], structural econometric [AGS20a; AGS20b] models. A different question asks for the effect of perturbing the *location* of $x_i$ spatially in the sample space $\mathcal{X}$ as $x_i + \Delta_i$ rather than its *probability weight*; this can also be answered with the IF. The Wasserstein gradient vector field $\nabla_x \psi_P = (\partial_{x_1} \psi_P, \ldots, \partial_{x_d} \psi_P) : \mathcal{X} \to \mathbb{R}^d$ of $\theta(P)$ describes the direction of transporting the mass of $P$ at $x$ in $\mathcal{X} \subset \mathbb{R}^d$ with the greatest influence on the value of $\theta(P)$. Furthermore, for any transport perturbation $\boldsymbol{v} \in L^2(P)^d$ of $P$, the effect on $\theta(P)$ is $D\theta_P[\boldsymbol{v}] = \int \langle \boldsymbol{v}, \nabla_x \psi_P \rangle_{\mathbb{R}^d} \, \mathrm{d}P$. See [Vil03, ch8] for the details of optimal transport theory, and e.g., [Mad+18; SND18] for applications.

A classical use of the IF is to approximate the distribution of an estimator [Cha92; New94; FS19], useful for constructing confidence intervals and statistical tests. From (1.1), the variance of $\hat{\theta}_n$ is approximately $\int \psi^2 \, \mathrm{d}P/n$ and the distribution is approximately Normal by the CLT. Beyond insights into the distribution of a given estimator, the IF is a key ingredient for the construction of estimators that achieve the efficiency bound, notably in semiparametric problems like causal inference. Several techniques are well known, all start with preliminary estimators $\check{\theta}$ of $\theta$ and $\hat{\psi}_P$ of $\psi_P$, which are combined to construct a better estimator of $\theta$. The one-step adaptive estimator of Bickel [Bic75] given by $\hat{\theta} := \check{\theta} + \sum_{i=1}^{n} \hat{\psi}_P(X_i)/n$ estimates and removes the bias in $\check{\theta}$ with the sample average of $\hat{\psi}_P$. The IF is required to construct the targeted likelihood of van der Laan [VR06; Cho+24], and Neyman orthogonal estimating equations of Chernozhukov [Che+18; Che+22], see also the reviews [Hin+22; Ken24].

Recently, the IF is used extensively to study robustness of ML models [Bre+19; Guo+20; Bae+22; Sch+23] and the sensitivity of econometric estimands $\theta$ to misspecification of structural properties of $P$ and other modeling choices reflected in $\theta(P)$ [AGS17; Muk18; Muk19; CC23]. For example, with censored data, the mean $\theta_1$ is not identified, but there is *a set* of values for $\theta_1$ compatible with the true $P$ and the observed data that can be estimated [HM95; Sem20; Sem25]; an MDP model may assume Gumbel payoff shocks [Rus87] or an ecological model of population dynamics may posit logistic propensity scores [Cat+00], and the IF can be used to construct bounds for the estimated MDP parameters and population size that are robust to the parametric assumptions [Muk21].

How does one estimate the IF? In Example 1.1, estimating $\psi_1$ amounts to estimating the mean $\theta_1$; by contrast, estimating $\psi_0$ requires estimating the density $f$; estimating $\psi_q$ requires estimating $\theta_q$ and $f(\theta_q)$. For a general functional $\theta(P)$, one first derives the analytic form of the IF. In parametric models, the IF is the normalized derivative of the log-density, i.e., the score. In nonparametric models, with significant mathematical subtlety, the problem can be reduced to the parametric case. For a path $t \mapsto P_t$, i.e., a parametric submodel with parameter $t \in \mathbb{R}$ and score function $g \in L_0^2(P)$, the pathwise derivative $d\theta(P_t)/dt$ is computed. Then one hopes to express it as $\int g \cdot \phi \, dP$ to match the representation of Riesz' theorem; the function $\phi$ is a candidate for $\psi_P$. Given the analytic form of $\psi_P$, as in Example 1.1, the unknown components can be estimated. Deriving the IF is highly idiosyncratic to the functional at hand and can be challenging, akin to solving a differential equation, and often constitutes a significant contribution. Several techniques are described in the literature e.g., [Cha87; New90; Jor93], see also the reviews [Hin+22; Ken24]. This task can be time-consuming, requires familiarity with functional analysis, and often highly specialized technical knowledge.

**Prior works** It is widely documented in recent literature that the analytic derivation of the IF poses an obstruction to the adoption of IF-based methods [Fra+15; LCL15; CLV19; Hin+22; Ken24; JWZ22] and that replacing this derivation with *automated* estimation would be a useful contribution. Automated estimation has been explored in [Fra+15; CLV19; JWZ22] based on the von Mises representation of the IF, explained in Theorem 2.1. Specifically, these works show how to estimate the value of the IF at a fixed point $z \in \mathcal{X}$ as $\hat{\psi}_P(z) = \lim_{\delta \to 0} d\theta(\hat{P}_t^{\delta,z})/dt$, by constructing a special perturbation $\hat{P}_t^{\delta,z}$ of $\hat{P}$, that depends on the chosen point $z$ and bandwidth $\delta$, so that the pathwise derivative of $\theta$ along this perturbation approximates $\psi_P(z)$. In practice, $\delta$ is fixed and the derivative is computed as a finite difference; computation can be ill-conditioned due to the nature of the required perturbations.

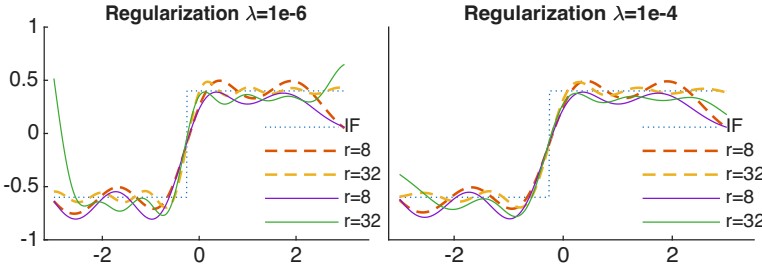

Figure 1: IF $\psi_q$ of quantile $\theta_q$, regularized oracle surrogate $\psi_\lambda^r$ (dashed) and estimate $\hat{\psi}_{\lambda,n}^r$ (solid).

**Contribution** We propose estimating the entire IF $\psi_P$ as

$$\hat{\psi}_\lambda^r(x) := \sum_{j=1}^r \frac{1}{1 + 2\lambda/\hat{\sigma}_j} \left[ \frac{d}{dt} \theta(\hat{f}_t^j) \right]_{|t=0} \hat{e}_j(x),$$

where $\lambda \geq 0$ is the regularization loading for controlling the nonparametric bias-variance trade-off, and $r$ is the rank of the approximation for controlling the number of eigenvectors of the Gram matrix and pathwise derivatives of $\theta$ one has to compute (e.g., $r = 16$) to estimate *the entire IF*. See Figure 1. Furthermore, $(e_j)_{j \geq 1}$ is an orthonormal basis for $L_0^2(P)$ and $(\sigma_j)_{j \geq 1}$ is a decreasing to 0 sequence of scalars such that $(\sqrt{\sigma_j} e_j)_{j \geq 1}$ is an orthonormal basis for a Hilbert space of smooth approximating functions; perturbations $f_t^j$ of the density $f$ of $P$ are in the direction of $e_j$, e.g., $\hat{f}_t^j = [1 + t\hat{e}_j]\hat{f}$, leading to stable pathwise derivative computation.

Our estimator is based on a novel regularized representation of $\psi_P$ by the best approximation in a ball of a Hilbert space $H$, Theorem 3.3. We view this representation as our main result, because it is not trivial to apply rkHs methods to the IF, and our variational characterization lends itself to this task. We relate estimation of the IF to kernel Principal Component Analysis (PCA) by taking $H$ to be the rkHs of a positive definite kernel. This allows leveraging extensive learning theory and methods of numerical linear algebra to analyze and compute our estimator. In the proposed implementation, both $e_j$ and $\sigma_j$ depend on $P$ and are estimated with a low-rank matrix eigenvalue and eigenvector decomposition. We prove consistency and lay the foundation for studying convergence rates, statistical and computational efficiency, and their trade-offs in follow-up work. To our knowledge, this is the first data-driven functional estimator of the IF, the first estimator with convergence guarantees in $L^2$ and rkHs norms, and the first work to show how to apply rkHs methods to the estimation of the IF.

**Paper organization** Section 2 reviews the classical von Mises formula, which is the baseline for our spectral representation and proposed estimator $\hat{\psi}_\lambda^r$. Section 3 presents our variation representation, regularization, and the solution of the resulting approximation problem. Section 4 implements the approximation with an rkHs, applies the Nyström method for integral operators to estimate the regularized surrogate and proves consistency of the estimator. The Appendix contains simulation experiments and deferred technical details.

**Notation** We write IF for influence function; $\mathcal{P}$ is a set of probability measures on a sample space $x \in \mathcal{X} \subset \mathbb{R}^d$; $X_{1:n} := (X_1, \ldots, X_n)$ is a sample; $\theta : \mathcal{P} \to \mathbb{R}$ denotes a functional; $L^2(P)$ is the space of functions $\phi : \mathcal{X} \to \mathbb{R}$ with $\int \phi^2 \, dP < \infty$ and $L_0^2(P)$ is the subspace with $\int \phi \, dP = 0$. The derivative of $\theta$ with respect to $P$ at $P$ is denoted by $D\theta_P : L_0^2(P) \to \mathbb{R}$ and maps directions of perturbing $P$ into infinitesimal changes in $\theta(P)$. $\mathbb{1}_A$ denotes the indicator of set $A$; $\mathscr{L}^d$ denotes Lebesgue measure on $\mathbb{R}^d$.

## 2 von Mises Formula

The following calculation extends those of [Mis47; IN22] and allows evaluation of the IF $\psi_P$ at a point $x \in \mathcal{X}$ via evaluations of the functional $\theta(P)$ on certain perturbations of $P$. A useful analogy is to think of computing partial derivatives of a multivariate function of $\mathbb{R}^2$ or $\mathbb{R}^3$ to evaluate the gradient vector.

THEOREM 2.1 [von Mises formula]. *Let $\theta : \mathcal{P} \to \mathbb{R}$ be a pathwise differentiable functional on a nonparametric model $\mathcal{P}$ with IFs $\psi_P \in L_0^2(P)$ for $P \in \mathcal{P}$. Suppose $P = f \cdot \mathscr{L}^d$ is a.c. with a continuous density $f$. Let $K$ be a bounded probability density with support in $\{|x| \leq 1\} \subset \mathbb{R}^d$. Define the dilated kernels by $K^\delta(x) := \delta^{-d} K(\delta^{-1} x)$ for $\delta > 0$. Translate to the location of approximation $z \in \mathbb{R}^d$ and control the likelihood ratio with $f$ via a cutoff as $K^{\delta,z}(x) := cK^\delta(z-x) \cdot \mathbb{1}_{\{f > \delta\}}(x)$ with $c^{-1} := \int_{\{f > \delta\}} K^\delta(z-x) \, dx$. For small $\delta$ and $z \in \{f > \delta\}$, consider the family, indexed by the bandwidth $\delta$, of paths $\{P_t^{\delta,z}\}_{-\epsilon < t \leq 1}$ with parameter $t$ and density $f_t^{\delta,z}(x) := (1-t)f(x) + tK^{\delta,z}(x)$. Note these paths perturb the measure $P$ toward the point-mass distribution at $z \in \mathcal{X}$, regularized via the approximation to the identity $K^\delta$. Then the following IF formula holds:*

$$\psi_P(z) = \lim_{\delta \to 0} \left[ \frac{d}{dt} \theta(P_t^{\delta,z}) \right]_{t=0} \qquad \text{for } P\text{-almost every } z \in \mathbb{R}^d. \tag{2.1}$$

*Proof.* We outline the proof as a way of reviewing pathwise differentiability and provide the details in the Appendix. The score function, i.e., the derivative of log-density, of the path $t \mapsto P_t^{\delta,z}$ at $P$ is

$$\phi_{\delta,z}(x) := \frac{d}{dt}_{|t=0} \log \left\{ f(x) + t \left[ K^\delta(z-x) - f(x) \right] \right\} = K^\delta(z-x)/f(x) - 1.$$

The score $\phi_{\delta,z}(x)$ is an $L_0^2(P)$ function that characterizes the infinitesimal change in the density at $x$ as $P$ is perturbed along the path $P_t^{\delta,z}$. Pathwise differentiability of $\theta$ at $P$ means that the derivative of the functional $\theta$ along the path $t \mapsto P_t^{\delta,z}$ exists and is a bounded linear functional of the score $\phi_{\delta,z}$. By Riesz' theorem [SS09, 4.5], [Dud18, 5.5.1] for bounded functionals on the Hilbert space $L_0^2(P)$, the derivative $D\theta_P[\phi_{\delta,z}]$ is given by the $L_0^2(P)$ inner product of the score $\phi_{\delta,z}$ with the IF $\psi_P$:

$$\frac{d}{dt}_{|t=0} \theta(P_t^{\delta,z}) = D\theta_P[\phi_{\delta,z}] = \int_{\mathrm{spt}P} \psi_P(x) \, K^\delta(z-x) \, dx = \left( \psi_P * K^\delta \right)(z).$$

The assumed properties of the mollification kernels $K^\delta$ ensure that it is an approximation to the identity [SS09, 3.2] in the sense that it converges as $\delta \to 0$ to the singular point-mass distribution in the integral pairing with a $L_{\mathrm{loc}}^1(\mathbb{R}^d)$ function. By the Lebesgue differentiation theorem [SS09, 3.3], [Dud18, 7.2] it follows that the convolution $(\psi_P * K^\delta)(z) \to \psi_P(z)$ converges as $\delta \to 0$ pointwise at the Lebesgue points of $\psi_P$ and therefore for $P$-almost every $z \in \mathbb{R}^d$. $\qquad\qquad\square$

Let's interpret Theorem 2.1. It says that to compute a single value of the IF, it is sufficient to compute the values of the functional $\theta$ along a certain perturbation to $P$. Specifically, $\psi_P(z)$ is the effect

on $\theta(P)$ of perturbing the $P$-weight of the outcome $z \in \mathcal{X}$. Recall the IF-methods described in Section 1 and note that these rely on this interpretation and the approximation of $\theta(P)$ by $\hat{\theta}$. Thus, provided with a device for computing the derivative $d\theta/dt$ in (2.1) *numerically*, one can numerically query $\psi_P$; indeed, [CLV19; JWZ22] use finite differences with a similar von Mises representation. Our regularity assumptions for this result are different from those in the literature, by employing Lebesgue differentiation we make no additional regularity assumption about $\psi_P \in L_0^2(P)$.

In statistical applications one typically requires the entire map $z \mapsto \psi(z)$ rather than a particular value $\psi(z)$. For example, to find influential data points for an estimate $\hat{\theta}$, one seeks the global maximum or level sets of $\psi$; for constructing a debiased estimator of $\theta$ one needs to integrate against $\psi$; to find influential data points in the Wasserstein sense, one needs to apply a differential operator to the gradient $\nabla_x \psi(x)$ and maximize the resulting score function. Therefore, in practice, formula (2.1) is used to evaluate *many* values of $\psi$ *simultaneously*. With this in mind, note that (2.1) requires a separate computation for each evaluation and that the required perturbations toward a point mass have been found numerically challenging [CLV19; JWZ22]. Also note that the regularization in (2.1) does not account for properties of the measure $P$ such as concentration or properties of the function $\psi$ such as smoothness. Furthermore, downstream tasks e.g., causal inference, require convergence rates in function norms for the estimator of the IF, while it is not known how to obtain these with (2.1). These observations suggest that (2.1) may not be numerically and statistically optimal for estimating the entire function $\psi$ or even isolated values of $\psi$. To address these, we propose a new representation.

# 3 Spectral von Mises representation

## 3.1 Exact representation

We begin by finding a variational representation of the IF in terms of pathwise derivatives of the functional. The following lemma is an immediate consequence of Riesz' theorem [Dud18, 5.5.1] and Cauchy-Schwarz inequality [Dud18, 5.1.4] and records in a suitable form the basic observation: the IF $\psi_P$ is the direction of perturbing $P$ with the most rapid variation in the functional $\theta(P)$ for the Fisher-Rao geometry (with the $L^2(P)$ metric tensor) of the model $\mathcal{P}$.

LEMMA 3.1. *Let $\theta$ be a pathwise differentiable functional on $\mathcal{P}$ with derivative $D\theta_P$ and IF $\psi_P$ for $P \in \mathcal{P}$. The IF is the unique solution to the following dual optimization problems:*

$$\psi_P = -\operatorname*{arg\,min}_{\phi \in L_0^2(P)}\left\{ D\theta_P[\phi] \ ; \ \|\phi\|_{L^2(P)} \leq 1 \right\}$$

$$\propto -\operatorname*{arg\,min}_{\phi \in L_0^2(P)}\left\{ D\theta_P[\phi] + \lambda_{\mathsf{p}}\|\phi\|_{L^2(P)}^2 \right\}, \qquad \lambda_{\mathsf{p}} > 0. \tag{3.1}$$

*The proportionality constant in (3.1) is 1 if and only if the penalty loading is $\lambda_{\mathsf{p}} = 1/2$.*

In (3.1) we used the duality between constrained and penalized optimization. For the exact representation of $\psi$, both the constraint and the penalty are in terms of the $L^2(P)$ norm i.e., the metric tensor of the Fisher-Rao distance on the tangent spaces of $\mathcal{P}$. In other words, the Fisher-Rao metric gives rise to the geometry where the IF is the gradient perturbation.

## 3.2 Regularized representation

The variational representation and the geometric interpretation of the IF suggest a strategy for constructing a regularized approximation of $\psi$ as follows. Suppose there is a function space $H \subset L_0^2(P)$ with a norm $\|\phi\|_H$ that quantifies a suitable notion of smoothness of functions $\phi \in H$. Suppose we wish to find the best approximation of $\psi$ in $H$ with a given degree of smoothness as measures by $\|\cdot\|_H$. For example, $H$ can be a Sobolev space. Then the projection $\psi_M$ of $\psi$ on the ball

$$B_M := \{\phi \ ; \ \|\phi\|_H \leq M\} \subset H \subset L_0^2(P)$$

of radius $M > 0$ is the desired approximation, and $M$ controls the degree of regularization. If $H$ is dense in $L_0^2(P)$, then we indeed obtain an approximation of $\psi$ by the projection $\psi_M$ that improves and converges to $\psi$ as $M \to \infty$.

LEMMA 3.2. *Let $\theta$ be a pathwise differentiable functional on $\mathcal{P}$ with derivative $D\theta_P$ and IF $\psi_P$ for $P \in \mathcal{P}$. Let $(H, \|\cdot\|_H)$ be a Hilbert space, densely contained in $L_0^2(P)$. Then the projection of the IF $\psi_P$ on the set $B_M$ is the unique solution to the following dual optimization problems:*

$$\psi_{P,M} := -\underset{\phi \in H}{\arg\min}\Big\{ D\theta_P[\phi] \; ; \; \|\phi\|_{L^2(P)} \leq 1 \text{ and } \|\phi\|_H \leq M \Big\}$$

$$= -\underset{\phi \in H}{\arg\min}\Big\{ D\theta_P[\phi] + 1/2\|\phi\|_{L^2(P)}^2 + \lambda_{\mathsf{r}}\|\phi\|_H^2 \Big\} \qquad =: \psi_{P,\lambda} \qquad (3.2)$$

*for some regularization loading $\lambda_{\mathsf{r}} = \lambda_{\mathsf{r}}(M) \geq 0$. Furthermore, $\psi_M \to \psi$ in $L_0^2(P)$ as $M \to \infty$ and, equivalently, $\psi_\lambda \to \psi$ in $L_0^2(P)$ as $\lambda_{\mathsf{r}} \to 0$.*

In (3.2) we again used the duality between constrained and penalized optimization. The regularized surrogate of $\psi$ is obtained by strengthening the metric on $\mathcal{P}$ and taking its gradient as the approximation of the IF.

## 3.3 Spectral representation

We obtained a regularized functional representation (3.2) of the IF in terms of the evaluation of the pathwise derivative of the parameter $\theta$:

$$D\theta_P[\phi] = \frac{d}{dt}_{|t=0} \theta(P_t^\phi), \quad \text{where} \quad \frac{d}{dt}_{|t=0} \log f_t^\phi = \phi, \quad P_t^\phi = f_t^\phi \cdot \mathscr{L}^d,$$

and the path $\{P_t^\phi\}_{0 \leq t < \epsilon}$ can be taken to be any regular parametric model with parameter $t$ and score function $\phi \in H$ at $P$. This representation of $\psi_P$ is rather implicit. But it is the correct representation because it emphasizes the geometry of the problem and lends to thinking about $\psi_P$ as a vector in an inner product space rather than a bag of numbers, one for each $x \in \mathcal{X}$. Returning to our analogy of computing the gradient of a multivariate function on $\mathbb{R}^3$, we make the main observation of this paper:

> **Main idea:** *The computationally fruitful way of thinking about partial derivatives of $\theta$ in $\mathcal{P}$ is not along perturbations toward a point mass at each $x \in \mathcal{X}$, but rather along the directions of an orthonormal basis on the tangent space $L_0^2(P)$.*

Equipped with this intuition, we solve the optimization problem (3.2) analytically to obtain an infinite Fourier series representation of the regularized IF $\psi_{P,\lambda}$. Our main result is the following

THEOREM 3.3 [Spectral von Mises formula]. *Suppose there exists an orthonormal basis of functions $\{\mathsf{e}_j : \mathcal{X} \to \mathbb{R}\}_{j \geq 1}$ for $L_0^2(P)$ and a decreasing to zero sequence of scalars $\{\sigma_j > 0\}_{j \geq 1}$ such that $\{\sqrt{\sigma_j}\mathsf{e}_j\}_{j \geq 1}$ is an orthonormal basis for the Hilbert space $H \subset L_0^2(P)$.*

*Let $S : L_0^2(P) \to H$ be the linear regularization operator given in diagonalized form by*

$$S[u](x) = \sum_{j=1}^\infty \sigma_j \langle u\,,\,\mathsf{e}_j \rangle_{L^2(P)} \mathsf{e}_j(x), \qquad u \in L_0^2(P) \qquad (3.3)$$

*and assume that its adjoint operator $S^* : H \to L_0^2(P)$ is the inclusion $S^*[v] = v$, so that the inner products of $H$ and $L_0^2(P)$ are related by*

$$\langle Su\,,\,v \rangle_H = \langle u\,,\,S^*v \rangle_{L^2(P)} = \langle u\,,\,v \rangle_{L^2(P)}, \qquad \text{for all} \quad u \in L_0^2(P),\, v \in H. \qquad (3.4)$$

*Let $\theta$ be a pathwise differentiable functional on $\mathcal{P}$ with derivative $D\theta_P$ and IF $\psi_P \in L_0^2(P)$ for $P \in \mathcal{P}$. Then the following representation holds in the norm of $H$:*

$$\psi_{P,\lambda}(x) = \lim_{r \to \infty} \sum_{j=1}^r \frac{1}{1 + 2\lambda/\sigma_j}\left[\frac{d}{dt}\theta(P_t^j)\right]_{|t=0} \mathsf{e}_j(x), \qquad (3.5)$$

*where, for each basis function $\mathsf{e}_j$, the path $t \mapsto P_t^j$ can be any regular perturbation of $P$ with the score function $\partial_t \log P_t^j = \mathsf{e}_j$.*

Let's interpret Theorem 3.3 and compare with Theorem 2.1. Formula (3.5) is similar to formula (2.1) in that it expresses the (regularized) IF $\psi_P$ in terms of pathwise derivatives of the functional $\theta(P)$ along certain regular perturbations $P_t$ of the measure $P$. So, in order to compute with formula (3.5), one requires the same numerical tools as for implementing formula (2.1). The main difference is in the choice of the perturbation directions $\partial_t \log P_t$ that are employed by the two representations: (i) In (2.1), the perturbation depends on the evaluation point $z \in \mathcal{X}$. By contrast, in (3.5) the scores $e_j$ are fixed, once the approximating space $H$ and the data distribution $P$ are fixed; and, once the derivatives $\{D\theta[e_j]\}_{j=1}^r$ are computed, an approximation to all values of $\psi_P$ are obtained. (ii) The directions of perturbation $e_j$ depend on the approximating function class $H$, which allows to adapt to the smoothness of $\psi$. (iii) In our proposed implementation, $H$ is taken to be a reproducing kernel Hilbert space (rkHs) of a positive semidefinite kernel $K$, and the scores $e_j$ can be interpreted as nonlinear principal components of the measure $P$, which allows to adapt to its effective dimension. (iv) Perturbation directions $e_j$ of $P$ are ordered by the magnitude of the corresponding multiplier sequence $\sigma_j$, which leads to a natural low-rank approximation for $\psi_\lambda$ by the first $r$ terms of the formula (3.5). (v) The directions $\{e_j\}_{j=1}^r$ for perturbing $P$ are well-behaved.

*Proof.* We outline the main ideas of the proof and provide the details in the Appendix. Let $J(\phi)$ denote the objective function in (3.2) and note that it is strictly convex, so that the first order conditions are necessary and sufficient for characterizing the function $\psi_\lambda \in H$ at which the unique minimum of $J$ is attained. For a direction $v \in H$, the Gateau derivative of the objective function evaluated at the candidate function $\phi \in H$ is

$$\partial_v J(\phi) = \langle v, \psi \rangle_{L^2(P)} + \langle v, \phi \rangle_{L^2(P)} + 2\lambda \langle v, \phi \rangle_H.$$

Applying the adjoint relationship (3.4) to express $L^2(P)$ inner products in terms of $H$ inner products, obtain $\partial_v J(\phi) = \langle v, S\psi \rangle_H + \langle v, S\phi \rangle_H + 2\lambda \langle v, \phi \rangle_H$, for all $\phi, v \in H$. It follows that the $H$ gradient of $J$ (the Riesz representer in the $H$ inner product) is given by $\delta_H J(\phi) = S\psi + S\phi + 2\lambda\phi$ and that the solution $\psi_\lambda$ of (3.2) is characterized by the first order condition $\delta_H J(\psi_\lambda) = 0 \in H$. Using the spectral resolution (3.3) of the smoothing operator $S$, note that the first order condition is equivalent to the system of equations

$$\sigma_j \langle \psi, e_j \rangle_{L^2(P)} + \sigma_j \langle \psi_\lambda, e_j \rangle_{L^2(P)} + 2\lambda \langle \psi_\lambda, e_j \rangle_{L^2(P)} = 0, \quad j \geq 1.$$

Solving for the Fourier coefficients $\langle \psi_\lambda, e_j \rangle_{L^2(P)}$ of the spectral resolution of $\psi_\lambda$ in $L^2(P)$, and applying Riesz' representation $D\theta[e_j] = \langle \psi, e_j \rangle_{L^2(P)}$, obtain formula (3.5). □

## 4  Regularization in the rkHs of a Mercer kernel

**rkHs notation** A linear space $H$ of functions $\phi : \mathcal{X} \to \mathbb{R}$ with an inner product $\phi, \varphi \mapsto \langle \phi, \varphi \rangle_H$ is an rkHs if (i) it is complete in the norm $\|\phi\|_H^2 = \langle \phi, \phi \rangle_H$ of the inner product, and (ii) for every $x \in \mathcal{X}$, the evaluation functional $\phi \mapsto \phi(x)$ is continuous in the norm of $H$. Convergence of a sequence $\phi_n \to \phi$ in the norm of $H$ implies pointwise, and often uniform or stronger, convergence.

According to Riesz' theorem [Dud18, t5.5.1], [SS09, t5.3], for every $x \in \mathcal{X}$, there exists a function $k_x \in H$ such that the evaluation functional has the representation $\phi(x) = \langle \phi, k_x \rangle_H$ for all $\phi \in H$. The function $K : \mathcal{X}^2 \to \mathbb{R}$ given by $K(x, y) := k_y(x) = \langle k_y, k_x \rangle_H$ is known as the reproducing kernel of $H$ and is guaranteed to be a symmetric and positive semidefinite map: the matrix $[K(x_i, x_j)]_{i,j=1}^n$ is strictly positive definite for all $n \geq 1$ and distinct $\{x_j\} \subset \mathcal{X}$; we call any such function a PSD kernel. We call $k_x(\cdot) = K(\cdot, x)$ the slice of $K$ at $x$. By linearity, any finite superposition of slices $x \mapsto \sum_{j=1}^n \alpha_j k_{x_j}(x)$, with $\{x_j\} \subset \mathcal{X}$ and $\{\alpha_j\} \subset \mathbb{R}$ and $n \in \mathbb{N}$, is in $H$. Conversely, according to Moore's theorem [PR16, t2.14], [CS08, ch4], any function $\phi \in H$ is a (possibly infinite) superposition of kernel slices; and, any PSD kernel $K$ generates an rkHs of superpositions for which it is the reproducing kernel. In practice, one picks a PSD kernel function and works with the associated rkHs somewhat implicitly because it's norm and inner product are not immediately obvious.

Widely used examples are the Gaussian kernel $K(x, y) = \exp(-\|x - y\|_2^2/\sigma^2)$, it produces a space of infinitely smooth functions; and the Laplacian kernel $K(x, y) = \exp(-\|x - y\|_2/\sigma)$, it produces the space of Sobolev functions with $(d + 1)/2$ square integrable derivatives. rkHs spaces are used extensively in numerical analysis [Wen04; FM15] and nonparametric estimation [Wai19; Bac24] due to their analytic and algebraic properties. We use a PSD kernel to form an estimator of the IF via the spectral representation (3.5). We assume the following properties, and call any such function a Mercer kernel:

ASSUMPTION 4.1 [Mercer kernel]. *(0) probability measure $P$ is absolutely continuous with compact support $\mathcal{X} \subset \mathbb{R}^d$; (i) $K : \mathcal{X}^2 \to \mathbb{R}$ is a PSD function; (ii) $K$ is continuous, bounded, so that its rkHs $H \subset L^2(P)$ and with the bound $\kappa := \max_{x \in \mathcal{X}} K(x, x) < +\infty$; (iii) for every $y \in \mathcal{X}$, $\int_{\mathcal{X}} k_y \, \mathrm{d}P = 0$ so that $H \subset L_0^2(P)$; (iv) $H$ is universal in the sense that it is dense in $L_0^2(P)$.*

In particular, the Gaussian and Laplacian kernels are universal; see [SFL11] [CS08, ch4.6]. The normalization property (ii) can be imposed on any PSD kernel $K$ via one step of the Cholesky algorithm $\tilde{K}(x, y) := K(x, y) - \int_{\mathcal{X}} k_x \, \mathrm{d}P \int_{\mathcal{X}} k_y \, \mathrm{d}P / \int_{\mathcal{X}^2} K \, \mathrm{d}P \mathrm{d}P$ is also PSD [PR16, ch4].

## 4.1 Spectral basis

For a Mercer kernel $K$, consider the integral operator $S_K : L_0^2(P) \to H \subset L_0^2(P)$ with signature $K$:

$$S_K[\phi](x) = \int_{\mathcal{X}} K(x, y)\phi(y) \, \mathrm{d}P(y), \qquad \phi \in L_0^2(P). \tag{4.1}$$

Under Assumption 4.1, $K \in L^2(P \otimes P)$ and $S$ bounded on $L_0^2(P)$ by the Cauchy-Schwarz inequality. Operator $S$ can be thought of as a continuous superposition of slices $k_y$ of the kernel, and the range of $S$ is properly contained in $H$. Note that the range of $S$ is the rkHs of the PSD function $\int k_x k_y dP$, in particular, it depends on the measure $P$ [PR16, ch11].

THEOREM 4.2 [Mercer]. *Let $K$ be a Mercer kernel on a compact sample space $\mathcal{X}$ and $P$ be a probability measure supported on $\mathcal{X}$. Then there is an orthonormal sequence $\{\mathsf{e}_j : \mathcal{X} \to \mathbb{R}\}_{j \geq 1}$ of continuous $L_0^2(P)$ eigenfunctions of the integral operator $S$ with signature $K$ defined in (4.1), and a corresponding decreasing to zero sequence of eigenvalues, repeated according to multiplicity, $\{\sigma_j > 0\}_{j \geq 1}$ such that: (i) $\{\mathsf{e}_j\}_{j \geq 1}$ is an orthonormal basis for $L_0^2(P)$; (ii) $\{\sqrt{\sigma_j}\mathsf{e}_j\}_{j \geq 1}$ is an orthonormal basis for $H$; (iii) $S$ has the diagonalization (3.3), and the adjoint $S^* : H \to L_0^2(P)$ is the inclusion operator $S^*\phi = \phi$, in particular, relationship (3.4) holds between the $L_0^2(P)$ and $H$ inner products.*

See [SS09, 4.6], [FM09] for background on integral operators and [SS12; Sun05] for extensions of Mercer's theorem to noncompact domains. In particular, Theorem 4.2 holds for the Gaussian kernel on $\mathbb{R}^d$ and a measure $P = \rho \cdot \mathscr{L}^d$ with $\rho \in L^2(\mathbb{R}^d)$, as shown in [Sun05, s4].

**Example 4.3.** For the Gaussian kernel $K(x, y) = \exp(-\epsilon^2|x - y|^2)$ on $\mathbb{R}$ and the centered Gaussian weight distribution $\rho(x) = \alpha \exp(-\alpha^2 x^2)/\sqrt{\pi}$, the Mercer basis is given by:

$$\mathsf{e}_j(x) = \gamma_j e^{-\delta^2 x^2} H_{j-1}(\alpha\beta x), \quad \sigma_j = \sqrt{\frac{\alpha^2}{\alpha^2 + \delta^2 + \epsilon^2}} \left[\frac{\alpha^2}{\alpha^2 + \delta^2 + \epsilon^2}\right]^{j-1}, \quad j \geq 1, \ x \in \mathbb{R},$$

where $H_j$ is the Hermite polynomial of degree $j$, and constants $\beta = (1 + [2\epsilon/\alpha]^2)^{1/4}$, $\gamma_j = (\beta/2^{j-1}\Gamma(j))^{1/2}$, $\delta^2 = \alpha^2(\beta^2 - 1)/2$ are defined in terms of the shape parameters $\epsilon$ and $\alpha$. In particular, the eigenvalues $\sigma_j$ decay exponentially, so that only the first few terms in (3.5) capture most of the variation when $\psi_P$ is smooth. Both $K$ and $\rho$ can be extended to $\mathbb{R}^d$ as tensor products.

## 4.2 Nyström method for integral operators

In order to estimate the IF with a given Mercer kernel $K$ via the spectral representation (3.5), the leading eigenvalues $\{\sigma_j\}_{j=1}^r$ and the corresponding eigenfunctions $\{\mathsf{e}_j\}_{j=1}^r$ of the integral operator $S_K$ are required. If $P$ is known, these can be computed numerically [FM15, 12.2.2]. If $P$ is unknown and only a random sample from $P$ is available, these must be estimated statistically. The Nyström method for approximating the eigendecomposition of the integral operator (4.1) is to discretize the integral with the empirical sum, reducing to an eigendecomposition of the empirical Gram matrix $\boldsymbol{K}_n := [K(X_i, X_j)/n]_{i,j=1}^n$. This is essentially the well-studied kernel PCA problem to estimate the main nonlinear features of $P$ [Mik+98; ZB05; Sha+05; RBD10; SS22b; SS22a]. A closely related problem is the functional PCA [Bos00], and the low-rank Gaussian process approximation [VV+08; BRV19; BRV20; SS20]. We find the exposition in [RBD10] particularly lucid and follow it closely.

LEMMA 4.4 [Hilbert space LLN]. *Let $K$ be a Mercer kernel on $(\mathcal{X}, P)$ generating the rkHs $H$ and let $X_1, \ldots, X_n$ be an i.i.d. sample from $P$. Let the integral operators $T_H, T_n : H \to H$ be defined by*

$$T_H[\phi](x) := \int_{\mathcal{X}} \langle \phi , k_y \rangle_H K(x, y) \, dP(y), \quad T_n[\phi](x) := \frac{1}{n} \sum_{j=1}^n \langle \phi , k_{X_i} \rangle_H K(x, X_i), \quad \phi \in H.$$

*Then $T_n \to T_H$ as $n \to \infty$ in the Hilbert-Schmidt norm in probability, and, for each $n \geq 1$,*

$$\|T_H - T_n\|_{\mathsf{HS}} \lesssim \frac{\kappa \sqrt{\tau}}{\sqrt{n}} \tag{4.2}$$

*with probability at least $1 - 2e^{-\tau}$.*

Note that $T_n$ is the empirical analogue of $T_H$ obtained by replacing the continuous integral with respect to $P$ by the discrete integral with respect to the empirical distribution of a random sample from $P$. By appealing to a suitable law of large numbers or concentration inequality, it follows that $T_n$ is consistent for $T_H$. Furthermore, $T_H$ is related to $S_K$, whereas $T_n$ is related to $\boldsymbol{K}_n$, providing a link between the continuous operator $S_K$ and the matrix $\boldsymbol{K}_n$ that is otherwise not immediately clear.

Recall that $S^*$ is the inclusion of $H$ into $L_0^2(P)$, and note that the operators $T_H = SS^*$ and $T_K := S^*S$ are essentially the same operator with the same action on all functions $\phi \in H$, differing only in the domain of definition and the embedding space of the range. In particular, the eigenvalues of $T_K$, $S$ and $T_H$ are exactly the same and the eigenfunctions of $T_K$ and $T_H$ are related by the inclusion (resp. regularization) operators $S^*$ (resp. $S$), in other words are the same functions but viewed as elements of $L_0^2(P)$ and $H$ respectively.

For the empirical analogues, the finite-rank operator $T_n$ and the Gram matrix $\boldsymbol{K}_n$ are similarly related via the Nyström restriction operator $R_n : H \to \mathbb{R}^n$ given by $R_n[\phi] = (\phi(X_j))_{j=1}^n$. Its adjoint $R_n^* : \mathbb{R}^n \to H$ is the Nyström extension operator given by $R_n^*[\boldsymbol{y}] = \sum_{j=1}^n y^j k_{X_j}/n$ for $\boldsymbol{y} = (y^1, \ldots, y^n) \in \mathbb{R}^n$ endowed with the inner product $\langle \boldsymbol{x} , \boldsymbol{y} \rangle_n := \sum_{j=1}^n x^j y^j/n$ . With this notation, we have $T_n = R_n^* R_n$ and $\boldsymbol{K}_n = R_n R_n^*$, from which it follows that $T_n$ and $\boldsymbol{K}_n$ have the same nonzero eigenvalues and the corresponding eigenvectors are related via the restriction (resp. extension) operators $R_n$ (resp. $R_n^*$).

The next result is an application of the perturbation bound of [Kat87], see also [RBD10], to infer consistency of the spectrum of $\boldsymbol{K}_n$ for the spectrum of $T_K := S^*S$.

LEMMA 4.5 [Consistency of eigenvalues]. *Let $K$ be a Mercer kernel on $(\mathcal{X}, P)$ generating the rkHs $H$ and let $X_1, \ldots, X_n$ be an i.i.d. sample from $P$. Let the integral operator $T_K : L_0^2(P) \to L_0^2(P)$ and the empirical Gram matrix multiplication operator $\boldsymbol{K}_n : \mathbb{R}^n \to \mathbb{R}^n$ be given by*

$$T_K[\phi](x) = \int_{\mathcal{X}} K(x, y)\phi(y) \, dP(y), \quad \boldsymbol{K}_n[\boldsymbol{y}] = \left[\frac{1}{n} K(X_i, X_j)\right]_{i,j=1}^n \boldsymbol{y}, \quad \phi \in L_0^2(P), \, \boldsymbol{y} \in \mathbb{R}^n.$$

*Let $\{\sigma_j\}_{j \geq 1}$ be the decreasing enumeration of the eigenvalues of $T_K$, repeated according to the multiplicity, and let $\{\hat{\sigma}_j\}_{j \geq 1}$ denote the analogous enumeration of the eigenvalues of $\boldsymbol{K}_n$, extended by zero. Then $\hat{\sigma}_j \to \sigma_j$ as $n \to \infty$ uniformly in probability, and, for each $n \geq 1$,*

$$\sup_{j \geq 1} |\sigma_j - \hat{\sigma}_j| \leq \|T_K - \boldsymbol{K}_n\|_{\mathsf{HS}} \lesssim \frac{\kappa \sqrt{\tau}}{\sqrt{n}}, \tag{4.3}$$

*and $\sum_{j \geq 1}(\sigma_j - \hat{\sigma}_j)^2 \leq \|T_K - \boldsymbol{K}_n\|_{\mathsf{HS}}^2 \lesssim \frac{\kappa^2 \tau}{n}$ and $\left|\sum_{j \geq 1}(\sigma_j - \hat{\sigma}_j)\right| = \left|\mathsf{tr}(T_H) - \mathsf{tr}(T_n)\right| \lesssim \frac{\kappa \sqrt{\tau}}{\sqrt{n}}$ with probability at least $1 - 2e^{-\tau}$.*

Consistency of the empirical eigenfunctions is framed in terms of projections to accommodate the nonuniqueness of an orthonormal basis corresponding to eigenvalues with multiplicity. For $N \in \mathbb{N}$, let $r(N)$ denote the total number of eigenvalues, accounting for the multiplicity, corresponding to the leading $N$ distinct eigenvalues $\sigma_{r(1)} > \ldots > \sigma_{r(N)}$ and let $\sigma_{r(N)+1}$ be the next largest distinct eigenvalue of $T_K$. Let $H_N := \mathsf{span}\{e_1, \ldots, e_{r(N)}\} \subset H \subset L_0^2(P)$ be the eigenspace of the $N$ leading distinct eigenvalues, and let

$$P_N : H \to H_N, \quad P_N[\phi] := \sum_{j=1}^{r(N)} \langle \phi , \sqrt{\sigma_j} e_j \rangle_H \sqrt{\sigma_j} e_j = \sum_{j=1}^{r(N)} \langle \phi , e_j \rangle_{L^2(P)} e_j, \quad \phi \in H$$

be its spectral projection. The following perturbation bound was used by [ZB05; KG00]: if $\hat{T}_K$ is a finite-rank estimate of the operator $T_K$ with precision on the order of the $N$th spectral gap with $\|T_K - \hat{T}_K\|_{\mathsf{op}} \le [\sigma_{r(N)} - \sigma_{r(N)+1}]/4$, then the eigenspaces $H_N$ of $T_K$ and $\hat{H}_N$ of the leading $r(N)$ eigenvalues of $\hat{T}_K$ must also be close with $\|P_N - P_{\hat{H}_N}\|_{\mathsf{op}} \le 2/[\sigma_{r(N)} - \sigma_{r(N)+1}]\|T_K - \hat{T}_K\|_{\mathsf{op}}$.

LEMMA 4.6 [Consistency of spectral projections]. *In the setting of Lemma 4.5, let $\mathbf{y}_1, \ldots, \mathbf{y}_n$ denote the orthonormal (for the scalar product $\langle \cdot\,,\, \cdot \rangle_n = \langle \cdot\,,\, \cdot \rangle_{\mathbb{R}^n}/n$ so that $\|\mathbf{y}_i\|_{\mathbb{R}^n} = \sqrt{n}$) eigenvectors of the empirical Gram matrix $\mathbf{K}_n$:*

$$\mathbf{K}_n[\mathbf{y}] = \sum_{i=1}^{n} \hat{\sigma}_i \langle \mathbf{y}\,,\, \mathbf{y}_i \rangle_n \mathbf{y}_i, \quad \mathbf{y} \in \mathbb{R}^n. \tag{4.4}$$

*Then, with $\mathbf{y}_i = (y_i^1, \ldots, y_i^n) \in \mathbb{R}^n$,*

$$\hat{e}_i(x) = \frac{1}{\hat{\sigma}_i} R_n^*[\mathbf{y}_i](x) = \frac{1}{\hat{\sigma}_i} \frac{1}{n} \sum_{j=1}^{n} y_i^j K(x, X_j), \quad i = 1, \ldots, n \tag{4.5}$$

*denote the eigenfunctions of the empirical integral operator $T_n$ (with normalization $\|\sqrt{\hat{\sigma}_i}\hat{e}_i\|_H \equiv 1$ and $\|\hat{e}_i\|_{L^2(P)} \approx 1$):*

$$T_n[\phi](x) = \sum_{i=1}^{n} \hat{\sigma}_i \langle \phi\,,\, \sqrt{\hat{\sigma}_i}\hat{e}_i \rangle_H \sqrt{\hat{\sigma}_i}\hat{e}_i(x), \quad \phi \in H.$$

*Let $N \in \mathbb{N}$ and $P_N$ (resp. $P_{\hat{H}_N}$) denote the spectral projection operator on the eigenspace of the leading $r(N)$ eigenvalues of $T_K$ (resp. $T_n$) and $I_H$ denote the identity on $H$. Then, for any $\psi \in H$,*

$$P_{\hat{H}_N}[\psi] = \sum_{j=1}^{r(N)} \hat{\sigma}_j \langle \psi\,,\, \hat{e}_j \rangle_H \hat{e}_j \to P_N[\psi] = \sum_{j=1}^{r(N)} \sigma_j \langle \psi\,,\, e_j \rangle_H e_j, \quad as\ n \to \infty$$

*in $H$ in probability, and, if $n \ge 128\kappa^2\tau/[\sigma_N - \sigma_{N+1}]^2$, then*

$$\sum_{j=1}^{r(N)} \|(I_H - P_N)\sqrt{\hat{\sigma}_j}\hat{e}_j\|_H^2 + \sum_{j=r(N)+1}^{n} \|P_N\sqrt{\hat{\sigma}_j}\hat{e}_j\|_H^2 \le \frac{32\kappa^2\tau}{(\sigma_{r(N)} - \sigma_{r(N)+1})^2 n}$$

*with probability at least $1 - 2e^{-\tau}$.*

## 4.3 Kernel von Mises estimator

Consistency of the estimated basis, via a continuous mapping theorem, implies consistency of the pathwise derivatives of $\theta$ evaluated along the estimated perturbation directions. The combination of the estimated basis and the estimated projection coefficients results in a consistent plug-in estimate of the regularized surrogate IF. We conclude with the following summary of our results:

THEOREM 4.7 [consistency of the kernel von Mises estimator]. *Let $K$ be a Mercer kernel on $(\mathcal{X}, P)$ generating the rkHs $H$ and let $X_1, \ldots, X_n$ be an i.i.d. sample from $P$. Let $\theta$ be a pathwise differentiable functional on $\mathcal{P}$ with derivative $D\theta_P$ and influence function $\psi_P \in L_0^2(P)$ for $P \in \mathcal{P}$. Assume that $D\theta_P$, equivalently $\psi_P$, are continuous in $P \in \mathcal{P}$ in an appropriate sense and that $\hat{f}$ is a consistent estimator of the density of $P$ in a compatible notion of convergence. Let $\{\hat{e}_j\}$ and $\{\hat{\sigma}_j\}$ be the Nyströrm estimators of the eigenfunctions and eigenvalues of $T_K$. For a fixed rank $1 \le r \le n$ and regularization loading $\lambda \ge 0$, let*

$$\psi_\lambda^r(x) := \sum_{j=1}^{r} \frac{D\theta_P[e_j]}{1 + 2\lambda/\sigma_j} e_j(x), \quad \hat{\psi}_\lambda^r(x) := \sum_{j=1}^{r} \frac{1}{1 + 2\lambda/\hat{\sigma}_j} \left[ \frac{d}{dt}\theta(\hat{f}_t^j) \right]_{|t=0} \hat{e}_j(x) \tag{4.6}$$

*denote the rank-r approximation of $\psi_P$ and its plug-in estimator obtained by replacing the unknown $\sigma_j, e_j, D\theta_P[e_j]$ with their estimates. The pathwise derivative can be computed along, e.g., the linear perturbation $\hat{f}_t^j = [1 + t\hat{e}_j]\hat{f}$ since the scores $e_j$ and $\hat{e}_j$ are bonded. Then $\|\hat{\psi}_\lambda^r - \psi_{P,\lambda}^r\|_H \to 0$ as $n \to \infty$ in probability and there exist sequences $r(n) \to \infty$ increasing and $\lambda(n) \to 0$ decreasing such that $\|\hat{\psi}_\lambda^r - \psi_P\|_{L^2(P)} \to 0$ as $n \to \infty$ in probability.*

**Limitations and future work**   Our spectral formula (3.5) and kernel implementation (4.6) aim to enable automation of methods based on asymptotic analysis and first-order techniques. These methods, whether carried out analytically or numerically, require theoretical justification and provide approximations that might be more or less accurate in a particular problem. For statistical inference, this is similar to the validity of bootstrap [Efr92]. It is hoped that the ideas provided here will allow leveraging efficient computation with PSD kernels [RCR15; Ste+20; Che+25], not considered here, for the applications described in the Introduction. However, further theoretical work is required: Downstream tasks, such as debiased machine learning [Che+18], make assumptions on the rate of convergence, e.g., $o(n^{-1/4})$, of the IF estimator. While we discuss some of the ingredients to study rates for our estimator, we only show consistency here and will investigate the rates in follow-up work. Furthermore, to unlock automation in earnest, further work is required to develop efficient estimates of the pathwise derivatives with automatic differentiation. Two strategies are possible: (i) introducing explicit weights $w_i = 1/n$ for each observation and perturbing this distribution; (ii) perturbing the *spatial location* of the observation $X_i$ and working with the duality of Wasserstein and Fisher-Rao tangent vectors.

**Acknowledgements**   I am grateful to Jan-Christian Hütter, Francesca Molinari, Kengo Kato, Kenji Fukumizu, Kyra Gan and Promit Ghosal for stimulating discussions, their support and interest. I thank the anonymous reviewers for their service, valuable feedback and suggestions to improve the exposition.

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

# A    Pathwise derivatives and von Mises formula

**Tangent space**  [KL76], [Bic+93, s3.2], [Vaa00, s25.3] and also [Vil03, s8.1.2] At each $P \in \mathcal{P}$ we consider perturbations to $P$ in $\mathcal{P}$ along one-dimensional parametric submodels $t \mapsto P_t \in \mathcal{P}$ with parameter $t \in [0, \epsilon)$ and $P = P_{t=0}$. These perturbations must be smooth and admit infinitesimal directions of perturbations; if we think of $\{P_t\}_t$ as a curve through $P$ in the space of probability measures, what is required is that it has a tangent vector at $P$. Let's assume that measures in our model are absolutely continuous $P = \rho \cdot \mathscr{L}^d$ with density function $\rho$ with respect to the Lebesgue measure $\mathscr{L}^d$ on $\mathbb{R}^d$. The direction of perturbation $P_t$ can then be identified with the time-derivative of the density along the curve $\partial_t \rho_t(x)$ for each point $x \in \mathcal{X} \subset \mathbb{R}^d$. The kinds of perturbations that are relevant for defining the influence function do not change the support of the distribution $P$, so $\partial_t \rho_t / \rho$ is well-defined, and it turns out to be more convenient mathematically to work with *the score function*

$$\phi(x) := \partial_{t|t=0} \log \rho_t(x), \qquad x \in \mathcal{X} \subset \mathbb{R}^d. \tag{A.1}$$

The score function $\phi : \mathcal{X} \to \mathbb{R}^2$ is the tangent vector to the curve $P_t$, the infinitesimal change in $P$ along the curve. The derivative in (A.1), need not hold pointwise, but rather in the Hellinger norm:

$$\lim_{t \to 0} \int_{\mathcal{X}} \left[ t^{-1} (\sqrt{\rho_t} - \sqrt{\rho}) - 2^{-1} \phi(x) \sqrt{\rho} \right]^2 d\mathscr{L}^d(x). \tag{A.2}$$

The existence of this limit implies that $\int \phi \, dP = 0$ and $\int \phi^2 \, dP < +\infty$. We denote the space of all such function by $L_0^2(P)$, indicating with the subscript that it is the subspace of $L^2(P)$ of all functions that have $P$-mean zero.

**Pathwise derivative**    A functional $\theta : \mathcal{P} \to \mathbb{R}$ is pathwise differentiable at $P$ (with respect to a collection of paths) if, (i) for a given regular path $P_t$ the composition $t \mapsto \theta(P_t)$ is a differentiable function from $[0, \epsilon)$ to $\mathbb{R}$ at time $t = 0$; and (ii) there is a bounded linear map $D\theta_P : L_0^2(P) \to \mathbb{R}$ such that

$$\lim_{t \to 0} t^{-1} [\theta(P_t) - \theta(P)] = D\theta_P[\phi] \tag{A.3}$$

for every score function $\phi \in L_0^2(P)$ and every admissible path $P_t$ with score $\phi$. The definitions of the score and pathwise derivative are those of Riemannian geometry that extends techniques of calculus to nonlinear spaces. Notions of smoothness are more nuanced in this infinite dimensional setting, see the penultimate paragraph in [Vil03, s3.2.3]

**Influence function**    By Reisz' representation theorem for Hilbert spaces [SS09, 4.5], [Dud18, 5.5.1], for the bounded linear functional $D\theta_P : L_0^2(P) \to \mathbb{R}$ there exists a fixed score $\psi_P \in L_0^2(P)$ such that the action of the derivative $D\theta_P$ on any score $v$ has the following representation in terms of the inner product:

$$D\theta_P[\phi] = \langle \phi, \psi_P \rangle_{L^2(P)}, \qquad \text{for all } \phi \in L_0^2(P). \tag{A.4}$$

The Riesz representer $\psi_P$ of the derivative functional of parameter $\theta(P)$ is known as the influence function. It has the useful geometric interpretation of the gradient score for parameter $\theta$, i.e., the direction of perturbation to $P$ such that the functional changes post rapidly:

$$D\theta_P[\phi] = \langle \phi, \psi_P \rangle_{L^2(P)} \leq \|\phi\|_{L^2(P)} \|\psi\|_{L^2(P)}, \quad \phi \in L_0^2(P) \tag{A.5}$$

where equality holds if and only if $v = c\psi$ by the Cauchy-Schwarz inequality. If we restrict the norm of the perturbation $\|\phi\|_{L^2(P)} \leq 1$ as in Lemma 3.1 and use linearity of $D\theta_P$, it follows that the unique maximum in (A.5) is achieved at the score $\psi_P / \|\psi_P\|_{L^2(P)}$ and the norm of the influence function $\|\psi_P\|_{L^2(P)}$ is the largest sensitivity of $\theta$ to a perturbation at $P$.

## A.1    Proof of Theorem 2.1

We begin with a score calculation for the original von Mises [Mis47] calculation with a point-mass perturbation. The example shows that this singular perturbation does not have a score function, hence the need to smooth out the point mass in general applications of this technique as in [IN22; CLV19].

**Example A.1 Score for the von Mises calculation.** Let $P$ and $\delta_x$ be a continuous distribution and a point mass on $\mathcal{X}$. Take the path $[0,1] \ni t \mapsto P_t = (1-t)P_t + t\delta_x$ for the calculations of von Mises and Huber [Hub72; Hub92]:

$$\partial_{t|t=0}\theta(P_t) = \lim_{t\to 0} t^{-1}[\theta(P_t) - \theta(P)] = \int_{\mathcal{X}} \psi_P \, \mathrm{d}[\delta_x - P] = \psi_P(x) \tag{A.6}$$

which cam be made rigorous if $\psi_P$ is, for instance, continuous at $x$. However, this perturbation to $P$ is not smooth in the sense of differentiability in quadratic mean (A.2). We compute the tangent vector to $P_t$ at $t = 1/3$ and $t = 0$. Take $\mu = P + \delta_x$ to be the dominating measure for the path, so that $f_t(z) = (1-t)\mathbb{1}_{\mathcal{X}\setminus x}(z) + t\mathbb{1}_{\{x\}}(z)$ is the Radon-Nikodym derivative at time $t$. The corresponding embedding of $P_t$ into the space of square roots of measures $H_2$ [BR07, c4] is $\sqrt{f_t}(z) = \sqrt{1-t}\mathbb{1}_{\mathcal{X}/x}(z) + \sqrt{t}\mathbb{1}_x(z)$. Note that the path in no longer linear in the embedding space. Also note that $d_{\mathsf{TV}}(P_t, P_{t+h}) = 2\sup_A |P_{t+h}[A] - P_t[A]| = 2h$ is continuous in $H_2$. For $t = 1/3$, the density $\sqrt{f_t}(z)$ can be differentiated pointwise for each $z \in \mathcal{X}$ to find the score function $\frac{1}{2}\phi_{\frac{1}{3}}(z)\sqrt{f_{\frac{1}{3}}}(z) = -\frac{1}{2}[\frac{2}{3}]^{-1/2}\mathbb{1}_{\mathcal{X}\setminus x}(z) + \frac{1}{2}[\frac{1}{3}]^{-1/2}\mathbb{1}_x(z)$ and verify differentiability in quadratic mean

$$\left\| t^{-1}[\sqrt{f_{\frac{1}{3}+t}} - \sqrt{f_{\frac{1}{3}}}] - \tfrac{1}{2}\phi_{\frac{1}{3}}(z)\sqrt{f_{\frac{1}{3}}}(z) \right\|^2_{H_2}$$

$$= \left\{ t^{-1}\left[ \sqrt{\tfrac{2}{3}-t} - \sqrt{\tfrac{2}{3}} \right] - (-1)\tfrac{1}{2}(\tfrac{2}{3})^{-1/2} \right\}^2 P[\mathcal{X}\setminus x]$$

$$+ \left\{ t^{-1}\left[ \sqrt{\tfrac{1}{3}+t} - \sqrt{\tfrac{1}{3}} \right] - \tfrac{1}{2}(\tfrac{1}{3})^{-1/2} \right\}^2 \delta_{\{x\}}[x]$$

$$= o(1) \qquad \text{as } t \to 0.$$

Repeating the calculation with $t = 0$, we note that the right derivative of $\sqrt{t}$ is infinite, so there is no score function with finite $\mu$-a.e. values that can satisfy (A.2). Consequently, the path $P_t$ is not smooth in the Hellinger norm and does not have a tangent vector at $t = 0$.

To remedy the lack of smoothness and extend the von Mises formula (A.6) to all pathwise differentiable functionals, the point-mass perturbations must be mollified.

LEMMA A.2 [Approximation to von Mises perturbation with a score]. *Suppose $K$ is a bounded probability density function on $\mathbb{R}^d$ with support in the unit ball $|x| \leq 1$. Then*

$$K^\delta(x) := \delta^{-d}K(\delta^{-1}x), \quad \delta > 0 \tag{A.7}$$

*is an* approximation to the identity *in the sense of [SS09, p109], that is*

(i) $\int_{\mathbb{R}^d} K^\delta(x)\, dx = 1$.

(ii) $|K^\delta(x)| \leq A\delta^{-d}$ *for all $\delta > 0$.*

(iii) $|K^\delta(x)| \leq A\delta/|x|^{d+1}$ *for all $\delta > 0$ and $x \in \mathbb{R}^d$.*

*Here $A$ is a constant independent of $\delta$.*

*Suppose $P_0$ is a probability measure that is absolutely continuous with respect to the Lebesgue measure $\mathscr{L}^d$ with a continuous density function $f_0$. Let*

$$K^{\delta,z}(x) := \left[ \int_{\{f_0 > \delta\}} K^\delta(z - x)\, \mathrm{d}x \right]^{-1} \mathbb{1}_{\{f_0 > \delta\}}(x)K^\delta(z - x), \tag{A.8}$$

*then for $z \in \{f_0 > 0\}$ we have $K^{\delta,z}(x) = K^\delta(z - x)$ for all sufficiently small $\delta > 0$ (which depend on $z$ that is fixed throughout). Furthermore,*

$$f_t^{\delta,z}(x) := (1-t)f_0(x) + tK^{\delta,z}(x) \tag{A.9}$$

*is a curve of probability densities with parameter $t$ in an interval around $0$, that is differentiable in quadratic mean (A.2) at $t = 0$ with the score function*

$$\phi_{\delta,z}(x) := \frac{d}{dt}_{|t=0} \log f_t^{\delta,z}(x) = \frac{K^{\delta,z}(x)}{f_0(x)} - 1. \tag{A.10}$$

*Proof.* The three properties of an approximation to the identity follow respectively from dilation invariance of Lebesgue integral, boundedness and compact support of the kernel $K$.

Fix a $z \in \{f_0 > 0\}$. By the continuity of $f_0$ there is a neighborhood $\mathcal{N}$ of $z$ such that $f_0$ is bounded away from zero on $\mathcal{N}$. For all $\delta > 0$ small enough, $x \mapsto K^\delta(z - x)$ is supported in $\mathcal{N}$ by bounded support and dilation construction, so that $K^\delta(z-x) \equiv K^{\delta,z}(x)$. Therefore for $t$ negative and close enough to 0, function $f_{t,\delta,z}$ is a well-defined probability density and its score functions

$$\phi_{t,\delta,z}(x) := \frac{d}{dt} \log f_t^{\delta,z}(x) = \frac{K^{\delta,z}(x) - f_0(x)}{f_t^{\delta,z}(x)} \tag{A.11}$$

are bounded in $x \in \mathcal{X}$. To check (A.2)

$$\int_{\mathcal{X}} \left[ \frac{\sqrt{f_{t,d,z}} - \sqrt{f_0}}{t} - \frac{1}{2} \phi_{\delta,z} \sqrt{f_0} \right]^2 \mathrm{d}x \to 0 \quad \text{as } t \to 0, \tag{A.12}$$

note that the map $t \mapsto \sqrt{f_{t,\delta,z}}(x)$ is continuously differentiable for each $x$ in a neighborhood $t \in (-\epsilon, \epsilon)$ of 0 with the derivative $\frac{1}{2} v_{t,\delta,z}(x) \sqrt{f_{t,\delta,z}(x)}$, therefore the problem is to justify the change of order of the limit $t \to 0$ and the integral $\int_{\mathcal{X}} dx$ in (A.12). By the fundamental theorem of calculus, we can write the difference quotient as

$$\sqrt{f_{0+ht}(x)} - \sqrt{f_0(x)} = \int_0^1 \frac{d}{dh} \sqrt{f_{0+ht}(x)} \, \mathrm{d}h = \int_0^1 \frac{1}{2} \phi_{0+ht}(x) \sqrt{f_{0+ht}} \cdot t \, \mathrm{d}h.$$

Therefore, by $(a - b)^2 \le 2a^2 + 2b^2$ and Cauchy-Schwarz inequality, we have the pointwise bound

$$\left[ \frac{\sqrt{f_{t,d,z}(x)} - \sqrt{f_0(x)}}{t} - \frac{1}{2} \phi_{0,\delta,z}(x) \sqrt{f_0(x)} \right]^2$$

$$\le 2 \left[ \int_0^1 \frac{1}{2} \phi_{ht,\delta,z}(x) \sqrt{f_{ht,\delta,z}} \, \mathrm{d}h \right]^2 + 2 \frac{1}{2} \phi_{\delta,z}(x)^2 f_0(x)$$

$$\le \int_0^1 \frac{1}{2} \phi_{ht,\delta,z}(x)^2 f_{0+ht} \, \mathrm{d}h + \phi_{\delta,z}(x)^2 f_0(x).$$

By the generalized Lebesgue dominated convergence theorem [Roy10, p89, t19], in order to conclude (A.12), it is sufficient to show that $\int_{\mathcal{X}} \int_0^1 \frac{1}{2} \phi_{ht,\delta,z}(x)^2 f_{0+ht} \, \mathrm{d}h \mathrm{d}x$ converges as $t \to 0$. By Fubini's theorem

$$\int_{\mathcal{X}} \int_0^1 \frac{1}{2} \phi_{ht,\delta,z}(x)^2 f_{0+ht} \, \mathrm{d}h \mathrm{d}x = \int_0^1 \int_{\mathcal{X}} \frac{1}{2} \phi_{ht,\delta,z}(x)^2 f_{0+ht} \, \mathrm{d}x \mathrm{d}h = \frac{1}{2} \int_0^1 I_{ht,\delta,z} \, \mathrm{d}h.$$

Since the scores (A.11) are bounded, the information matrix $I_{t,\delta,z}$ is continuous in $t$ at 0, and the above integral converges to $I_{0,\delta,z}$. $\qquad \square$

*Proof of Theorem 2.1.* By the pathwise differentiability of functional $\theta$, differentiability in quadratic mean of the path $t \to P_t^{\delta,z}$, and Riesz' representation we have

$$\frac{d}{dt}_{|t=0} \theta(P_t^{\delta,z}) = D\theta_P[\phi_{\delta,z}]$$

$$= \int_{\mathcal{X}} \psi_P(x) \, \phi_{\delta,z}(x) \, \mathrm{d}P.$$

Assume that $\psi_P(x) = \psi_P(x) \mathbb{1}_{\{f_0 > 0\}}(x)$. Below $P$ is fixed and we drop the subscript $P$ for convenience. Using the score $\phi_{\delta,z}$ computed in Lemma A.2 and the fact that $\psi$ has zero $P$-mean, have the expression for the pathwise derivative as the convolution of the influence function with the approximation to identity kernels:

$$\frac{d}{dt}_{|t=0} \theta(P_t^{\delta,z}) = \int \psi_P(x) \left[ K^\delta(z - x)/f_0(x) - 1 \right] \mathrm{d}P$$

$$= \int_{\mathrm{spt}P} \psi_P(x) \, K^\delta(z - x) \, \mathrm{d}x - 0$$

$$= (\psi_P * K^\delta)(z).$$

It suffices to show that for each $\alpha > 0$ and $M > 0$ the set

$$E_\alpha = \left\{ z \in \mathsf{spt} P \; ; \; \limsup_{\delta \to 0} \left| (\psi_P * K^\delta)(z) - \psi_P(z) \right| > 2\alpha \right\}$$

has zero Lebesgue measure, because then $E = \bigcup_{j=1}^\infty [E_{1/j} \cap \{|z| \le j\}]$ has zero measure by monotonicity, and the assertion (2.1) of the Theorem holds at all points $z \in E^c$. Thus, we may assume that $\psi_P$ has compact support and therefore belongs to $L^1(\mathbb{R}^d)$.

Because $K^\delta$ is a bounded probability density function, with support in $|x| \le \delta$ by the dilation construction (A.7), we can write

$$\left| (\psi_P * K_\delta)(z) - \psi_P(z) \right| = \left| \int_{\mathbb{R}^d} \left[ \psi_P(z - x) - \psi_P(z) \right] K_\delta(x) \, dx \right|$$

$$\le \int_{\mathbb{R}^d} \left| \psi_{P_0}(z - x) - \psi_{P_0}(z) \right| K_\delta(x) \, dx$$

$$\le \frac{c}{\delta^d} \int_{|x| \le \delta} \left| \psi_{P_0}(z - x) - \psi_{P_0}(z) \right| \, dx.$$

Fix $\alpha > 0$ and recall that continuous functions of compact support are dense in $L^1(\mathbb{R}^d)$ [SS09, p71], so that for each $\epsilon > 0$ we can choose a function $g$ with $\|\psi_P - g\|_{L^1(\mathbb{R}^d)} < \epsilon$. By the triangle inequality we can upper bound the expression above with

$$\frac{c}{\delta^d} \int_{|x| \le \delta} \left| \psi_P(z - x) - g(z - x) \right| \, dx + \frac{c}{\delta^d} \int_{|x| \le \delta} \left| g(z - x) - g(z) \right| \, dx + c' |g(z) - \psi_P(z)|.$$

By the continuity of $g$ it follows that

$$\lim_{\delta \to 0} \frac{c}{\delta^d} \int_{|x| \le \delta} \left| g(z - x) - g(z) \right| \, dx = 0, \qquad \text{for all } z.$$

We find that

$$\limsup_{\delta \to 0} \left| (\psi_P * K_\delta)(z) - \psi_P(z) \right| \le c' \left| \psi_P - g \right|^*(z) + c' |g(z) - \psi_P(z)|,$$

where the superscript $*$ indicates the Hardy-Littlewood maximal function:

$$f^*(x) := \sup_{B \ni x} \frac{1}{\mathscr{L}^d[B]} \int_B |f(y)| \, dy, \qquad \text{for } f \in L^1(\mathbb{R}^d), \quad x \in \mathbb{R}^d. \tag{A.13}$$

If we set

$$F_\alpha = \{ z \in \mathsf{spt} P \; ; \; \left| \psi_P - g \right|^*(z) > \alpha \} \quad \text{and} \quad G_\alpha = \{ z \in \mathsf{spt} P \; ; \; \left| \psi_P(z) - g(z) \right| > \alpha \}$$

then $E_\alpha \subset F_\alpha \cup G_\alpha$ by De Morgan's law since $E_\alpha^c \supset F_\alpha^c \cap G_\alpha^c$. Furthermore, by Chebyshev's inequality

$$\mathscr{L}^d[G_\alpha] \le \frac{1}{\alpha} \|\psi_{P_0} - g\|_{L^1(\mathbb{R}^d)},$$

and by the Hardy-Littlewood maximal inequality [SS09, p101]

$$\mathscr{L}^d[F_\alpha] \le \frac{3^d}{\alpha} \|\psi_{P_0} - g\|_{L^1(\mathbb{R}^d)}.$$

Recall that the function $g$ was chosen such that $\|\psi_{P_0} - g\|_{L^1(\mathbb{R}^d)} < \epsilon$, so that

$$\mathscr{L}^d[E_\alpha] \le c' \frac{3^d}{\alpha} \epsilon + c' \frac{1}{\alpha} \epsilon.$$

Since $\epsilon > 0$ is arbitrary, we conclude that $\mathscr{L}^d[E_\alpha] = 0$ and consequently $P[\bigcup_{j=1}^\infty E_{1/j}] = 0.$ $\qquad \square$

# B  Spectral representation

## B.1  Calculation for Lemma 3.1

The characterization of the influence function as a constrained optimizer was discussed in Appendix A around equation (A.5).

We verify the equivalence of the constrained problem and the penalized problem via an explicit calculation that is simple and instructive for the calculation of the spectral representation. Define the penalized objective function with penalty loading $\lambda_{\mathsf{pen}} > 0$:

$$J^{\backprime}(u) := D\theta_P[u] + \lambda_{\mathsf{pen}}\|u\|^2_{L^2(P)}, \quad u \in L^2_0(P). \tag{B.1}$$

Observer that $J^{\backprime}$ is strictly convex on $L^2_0(P)$ by the Cauchy-Schwarz inequality. By the strict convexity, the unique minimum of $J$ is attained at the tangent vector $u_0 \in L^2_0(P)$ where the derivative functional of $J^{\backprime}$ vanishes [Lue97]:

$$DJ^{\backprime}_{u_0}[v] = 0 \quad \text{for all } v \in L^2_0(P). \tag{B.2}$$

To compute the derivative of $J^{\backprime}$ at some $u$, fix a direction $v \in L^2_0(P)$ of perturbation and compute the difference quotient

$$\begin{aligned}
J^{\backprime}(u + \epsilon v) - J^{\backprime}(u) &= \left\{D\theta_P[u + \epsilon v] + \lambda_{\mathsf{pen}}\|u + \epsilon v\|^2_{2,P}\right\} - \left\{D\theta_P[u] + \lambda_{\mathsf{pen}}\|u\|^2_{2,P}\right\} \\
&= \left\{\langle u + \epsilon v,\, \psi\rangle_{2,P} + \lambda_{\mathsf{pen}}\langle u + \epsilon v,\, u + \epsilon v\rangle_{2,P}\right\} \\
&\qquad\qquad - \left\{\langle u,\, \psi\rangle_{2,P} + \lambda_{\mathsf{pen}}\langle u,\, u\rangle_{2,P}\right\} \\
&= \epsilon\left\{\langle v,\, \psi\rangle_{2,P} + 2\lambda_{\mathsf{pen}}\langle v,\, u\rangle_{2,P}\right\} + O(\epsilon^2).
\end{aligned} \tag{B.3}$$

We find that the gradient (Riesz representer) of the derivative functional of $J^{\backprime}$ at vector $u$ is

$$\nabla J^{\backprime}(u) = \psi + 2\lambda_{\mathsf{pen}}u. \tag{B.4}$$

Using Riesz' representation, the first order condition (B.2) becomes

$$\begin{aligned}
0 = DJ^{\backprime}_{u_0}[v] &= \big\langle v,\, \nabla J^{\backprime}(u)\big\rangle_{2,P} \\
&= \langle v,\, \psi_P + 2\lambda_{\mathsf{pen}}u_0\rangle_{2,P} \quad \text{for all } v \in L^2_0(P).
\end{aligned}$$

and conclude that $u_0 = -\psi/2\lambda_{\mathsf{pen}}$ is the minimizer of $J^{\backprime}$.

From this explicit solution to the penalized problem (3.1) we see that the direction of solution is always along the influence function, larger penalty loading $\lambda_{\mathsf{pen}}$ leads to solution with a smaller $L^2(P)$ norm, and $\lambda^*_{\mathsf{pen}} = 1/2$ uniquely identifies the influence function.

## B.2  Calculation for Lemma 3.2

We define the projection of the infuence function $\psi_P$ on the ball $B_M$ as the solution of the constrained optimization program. Define the penalized objective function $J^{\backprime\backprime}$ on $L^2_0(P)$ with the regularization loading $\lambda \geq 0$:

$$J^{\backprime\backprime}(u) := D\theta_P[u] + \lambda_{\mathsf{pen}}\|u\|^2_{2,P} + \lambda_{\mathsf{reg}}\|u\|^2_{L^2(P)}, \quad u \in L^2_0(P). \tag{B.5}$$

We observe that the constrained problem has the linear objective $D\theta_P[v]$, that the constraints are given by quadratic functionals and that the problem satisfies Slater's condition and that the strong convex duality holds.

The penalized objective $J^{\backprime\backprime}$ is strictly convex and the first order optimality condition

$$DJ^{\backprime\backprime}_{u_0}[v] = 0, \qquad \text{for all } v \in L^2 \tag{B.6}$$

is necessary and sufficient. Compute the difference quotient:

$$\begin{aligned}
&J^{\backprime\backprime}(u + \epsilon v) - J^{\backprime\backprime}(u) \\
&= \left\{D\theta_P[u + \epsilon v] + \lambda_{\mathsf{pen}}\|u + \epsilon v\|^2_{2,P} + \lambda_{\mathsf{reg}}\|u + \epsilon v\|^2_H\right\} \\
&\qquad\qquad - \left\{D\theta_P[u] + \lambda_{\mathsf{pen}}\|u\|^2_{2,P} + \lambda_{\mathsf{reg}}\|u\|^2_H\right\} \\
&= \left\{\langle u + \epsilon v,\, \psi\rangle_{2,P} + \lambda_{\mathsf{pen}}\langle u + \epsilon v,\, u + \epsilon v\rangle_{2,P} + \lambda_{\mathsf{reg}}\langle u + \epsilon v,\, u + \epsilon v\rangle_H\right\} \\
&\qquad\qquad - \left\{\langle u,\, \psi\rangle_{2,P} + \lambda_{\mathsf{pen}}\langle u,\, u\rangle_{2,P} + \lambda_{\mathsf{reg}}\langle u,\, u\rangle_H\right\} \\
&= \epsilon\left\{\langle v,\, \psi\rangle_{2,P} + 2\lambda_{\mathsf{pen}}\langle v,\, u\rangle_{2,P} + 2\lambda_{\mathsf{reg}}\langle v,\, u\rangle_H\right\} + O(\epsilon^2).
\end{aligned} \tag{B.7}$$

Take the limit as $\epsilon \to 0$ to obtain:

$$\partial_v J``(u) = \langle v \, , \, \tilde{\theta} \rangle_{2,P} + 2\lambda_{\mathsf{pen}} \langle v \, , \, u \rangle_{2,P} + 2\lambda_{\mathsf{reg}} \langle v \, , \, u \rangle_H. \tag{B.8}$$

From the first order condition, as $\lambda_{\mathsf{reg}} \to 0$, the optimal solution $u_0$ converges to that of the penalized but unregularized objective function $J`$.

## B.3   Proof of Theorem 3.3

First we check that the relationship (3.4) between the inner products of $L_0^2(P)$ and $H$ actually follows from the assumptions about the bases. Suppose $\{e_j\}$ and $\{\sqrt{\sigma}_j e_j\}$ are orthonormal bases (ONB) for $L_0^2(P)$ and $H$ respectively and the operator $S : L_0^2(P) \to H$ is defined by (3.3). From the definition of the adjoint $S^* : H \to L_0^2(P)$

$$\langle Se_j \, , \, e_i \rangle_H = \langle e_j \, , \, S^* e_i \rangle_{2,P} \qquad \text{all } i, j. \tag{B.9}$$

By the ONB assumption,

$$1 = \langle e_i \, , \, e_i \rangle_{2,P} = \langle \sqrt{\sigma}_i e_i \, , \, \sqrt{\sigma}_i e_i \rangle_H \qquad \text{all } i. \tag{B.10}$$

On the other hand, applying the eigenfunction property to (B.10) and using bilinearity of the inner product

$$\langle e_i \, , \, e_i \rangle_{2,P} = \langle \sigma_i e_i \, , \, e_i \rangle_H = \langle Se_i \, , \, e_i \rangle_H \qquad \text{all } i. \tag{B.11}$$

Similarly, we check for $i \neq j$,

$$0 = \langle \frac{1}{\sqrt{\sigma}_i} Se_i \, , \, \sqrt{\sigma}_j e_j \rangle_H = \frac{\sqrt{\sigma}_j}{\sqrt{\sigma}_i} \langle e_i \, , \, S^* e_j \rangle_{2,P} \qquad \text{all } i \neq j. \tag{B.12}$$

Since $\sigma_j / \sigma_i \neq 0$ and $\{e_i\}$ is complete, it follows that $e_j$ is an eigenfunction of $S^*$, and in the view of (B.11), the eigenvalue is 1 so that $S^*[e_j]$ must be equal to $e_j$. In other words, $S^*$ is the inclusion operator $H \to L_0^2(P)$.

Next, we use the adjoint relationship (B.9) and (3.4) in the expression (B.8) for the directional derivative of the objective function $J$:

$$\partial_v J(u) = \langle v \, , \, \psi_P \rangle_{2,P} + 2\lambda_{\mathsf{pen}} \langle v \, , \, u \rangle_{2,P} + 2\lambda_{\mathsf{reg}} \langle v \, , \, u \rangle_H \tag{B.13}$$

$$= \langle v \, , \, S\psi_P \rangle_H + 2\lambda_{\mathsf{pen}} \langle v \, , \, Su \rangle_H + 2\lambda_{\mathsf{reg}} \langle v \, , \, u \rangle_H. \tag{B.14}$$

It follows that the $H$ gradient (the representer in Riesz' representation for Hilbert spaces) of $DJ_u$ is given by:

$$\delta_H J(u) = S[\psi_P] + 2\lambda_{\mathsf{pen}} S[u] + 2\lambda_{\mathsf{reg}} u \tag{B.15}$$

$$= \sum_j \left\{ \sigma_j \langle \psi \, , \, e_j \rangle_{2,P} + 2\sigma_j \lambda_{\mathsf{pen}} \langle u \, , \, e_j \rangle_{2,P} + 2\lambda_{\mathsf{reg}} \langle u \, , \, e_j \rangle_{2,P} \right\} e_j. \tag{B.16}$$

With this expansion of the gradient $\delta_H J(u)$, the first order condition

$$\delta_H J(\psi_\lambda) = 0, \qquad \psi_\lambda \in H \tag{B.17}$$

of the penalized and regularized optimization program (3.2) becomes the follow system of equations:

$$0 = \sigma_j \langle \psi \, , \, e_j \rangle_{L^2(P)} + 2\sigma_j \lambda_{\mathsf{pen}} \langle \psi_\lambda \, , \, e_j \rangle_{L^2(P)} + 2\lambda_{\mathsf{reg}} \langle \psi_\lambda \, , \, e_j \rangle_{L^2(P)}, \quad j \geq 1. \tag{B.18}$$

Solving for the $L_0^2(P)$ Fourier coefficients of the optimal solution $\psi_\lambda$:

$$\langle \psi_\lambda \, , \, e_j \rangle_{L^2(P)} = \frac{\sigma_j}{2\sigma_j \lambda_{\mathsf{pen}} + 2\lambda_{\mathsf{reg}}} \langle \psi \, , \, e_j \rangle_{L^2(P)}, \quad j \geq 1. \tag{B.19}$$

Conclude that the optimal solution of (3.2) has the following Fourier series representation

$$\psi_{P,\lambda}(x) = \sum_{j=1}^{\infty} \frac{1}{2\lambda_{\mathsf{pen}} + 2\lambda_{\mathsf{reg}}/\sigma_j} \left[ \langle \psi_P \, , \, e_j \rangle_{L^2(P)} \right] e_j(x). \tag{B.20}$$

Observe that the sequence of $L^2(P)$ coefficients is shrunk toward zero by the eigenvalue sequence $\{\sigma_j\}$ and is in fact a valid sequence of coefficients for an element in $H$.

Finally, recall that $\langle \psi_P \, , \, e_j \rangle_{L^2(P)} = D\theta_P[e_j]$ and that the penalty loading should be $\lambda_{\mathsf{pen}} = 1/2$ for the correct scaling of the influence function from Lemma 3.1.

## C Nyström method

Our proofs of Lemmas 4.4, 4.5, 4.6 are modifications of [RBD10, Thm7, Prop10, Thm12].

### C.1 Proof of Lemma 4.4

Define the sequence of random operators $\xi_i : H \to H$ given by

$$\xi_i[\phi] = \langle \phi, \, k_{X_i} \rangle_H k_{X_i} - T_H[\phi], \quad \phi \in H, \ i = 1, \ldots, n. \tag{C.1}$$

We compute the norm of the continuous operator: for any orthonormal basis $\{\phi_j\}_{j \geq 1}$ of the rkHS $H$

$$
\begin{aligned}
\|T_H\|_{\mathsf{HS}}^2 &= \sum_{j \geq 1} \|T_H \phi_j\|_H^2 \\
&= \sum_{j \geq 1} \left\| \int_{\mathcal{X}} \phi_j(x) k_x \, \mathrm{d}P(x) \right\|_H^2 \\
&= \sum_{j \geq 1} \left\langle \int_{\mathcal{X}} \phi_j(x) k_x \, \mathrm{d}P(x), \, \int_{\mathcal{X}} \phi_j(x) k_x \, \mathrm{d}P(x) \right\rangle_H \\
&= \sum_{j \geq 1} \int_{\mathcal{X}} \int_{\mathcal{X}} \langle \phi_j(x) k_x, \, \phi_j(y) k_y \rangle_H \, \mathrm{d}P(x) \mathrm{d}P(y) \\
&= \sum_{j \geq 1} \int_{\mathcal{X}} \int_{\mathcal{X}} \phi_j(x) \phi_j(y) K(x,y) \, \mathrm{d}P(x) \mathrm{d}P(y) \\
&= \int_{\mathcal{X}} \int_{\mathcal{X}} \Big\{ \sum_{j \geq 1} \phi_j(x) \phi_j(y) \Big\} K(x,y) \, \mathrm{d}P(x) \mathrm{d}P(y) \\
&= \int_{\mathcal{X}} \int_{\mathcal{X}} \Big\{ K(x,y) \Big\} K(x,y) \, \mathrm{d}P(x) \mathrm{d}P(y) = \|K\|_{L^2(P \otimes P)}^2
\end{aligned}
$$

where we exchanged the Bochner integral with the inner product by Bochner integrability, used the reproducing property of the kernel, and the standard Mercer expansion of the kernel in the orthonormal basis that converges uniformly, exchanged the sum with the double integral by Fubini's .

Compute the Hilbert-Schmidt norm of the empirical operator, noting that $k_{X_i}$ is the eigenfunction of the rank-1 operator:

$$\|\xi_i\|_{\mathsf{HS}} \leq \|\phi(X_i) k_{X_i}\|_{\mathsf{HS}} + \|T_H\|_{\mathsf{HS}} \leq |K(X_i, X_i)| + \|K\|_{L^2(P \otimes P)} \leq 2\kappa.$$

This norm is an integrable real-valued random variable and therefore $\xi_i$ is Bochner integrable with the expectation

$$E[\xi_i] = \int_{\mathcal{X}} \langle \cdot, \, k_x \rangle k_x \, \mathrm{d}P(x) - T_H = 0.$$

By the strong law of large numbers for a random sequences in a separable Hilbert space (the space of Hilbert-Schmidt operators on $H$ in our case) [Bos00, Thm2.4]

$$\|T_n - T_H\|_{\mathsf{HS}} = \left\| \frac{1}{n} \sum_{i=1}^n \xi_i \right\|_{\mathsf{HS}} \to 0 \quad a.s.$$

Furthermore, applying the Hoeffding inequality for bounded (in norm, as verified above) random elements of a separable Hilbert space (the space of Hilbert-Schmidt operators on $H$) [Pin12], obtain

$$\left\| \frac{1}{n} \sum_{i=1}^n \xi_i \right\|_{\mathsf{HS}} \leq \frac{2\kappa\sqrt{2\tau}}{\sqrt{n}} \tag{C.2}$$

with probability at least $1 - 2e^{-\tau}$.

## C.2  Proof of Lemma 4.5

Applying [Kat87] to the empirical operator $B = T_n$ and the population counterpart operator $A = T_H$ defined on the separable rkHs $H$, for any nonnegative convex function $\Phi$ with $\Phi(0) = 0$:

$$\sum_{j \geq 1} \Phi(\hat{\sigma}_j - \sigma_j) \leq \sum_{j \geq 1} \Phi(\gamma_j)$$

where $\{\gamma_j\}_{j \geq 1}$ is an extended by zero enumeration of the eigenvalues of the random operator

$$B - A = T_n - T_H = \frac{1}{n} \sum_{i=1}^{n} \xi_i$$

defined in equation (C.1).

We apply [Kat87] with the choice $\Phi(s) = |s|^p$ for $p \geq 1$. In particular, with $p = 2$, this becomes

$$\sum_{j \geq 1} |\hat{\sigma}_j - \sigma_j|^2 \leq \sum_{j \geq 1} |\gamma_j|^2 = \|T_n - T_H\|_{\mathsf{HS}}^2 \leq \frac{(2\kappa)^2 2\tau}{n}$$

with probability at least $1 - 2e^{-\tau}$ from the bound (C.2).

Recalling that the sup norm is the limit of the $p$-norms:

$$\sup_{j \geq 1} |\hat{\sigma}_j - \sigma_j| = \lim_{p \to \infty} \left[ \sum_{j \geq 1} (\hat{\sigma}_j - \sigma_j)^p \right]^{\frac{1}{p}} \leq \sup_{p \to \infty} \left[ \sum_{j \geq 1} |\gamma_j^p| \right]^{\frac{1}{p}} = \sup_{j \geq 1} |\gamma_j|$$

$$= \|T_n - T_H\|_{\mathsf{op}} \leq \|T_n - T_H\|_{\mathsf{HS}} \leq \frac{2\kappa\sqrt{2\tau}}{\sqrt{n}}$$

with probability at least $1 - 2e^{-\tau}$ from the bound (C.2).

Given $\varepsilon > 0$, set $\varepsilon = \frac{2\kappa\sqrt{2\tau}}{\sqrt{n}}$ and solve for $\tau$ to obtain $\tau = n\varepsilon^2/2(2\kappa)^2$. Inverting the above finite sample concentration bound, find

$$P\left[ \sup_{j \geq 1} |\hat{\sigma}_j - \sigma_j| \geq \varepsilon \right] \leq 2e^{-n\varepsilon^2/2(2\kappa)^2} \to 0 \quad \text{as } n \to \infty.$$

For the bound on the difference of the traces, compute the trace of the empirical operator:

$$\sum_{j \geq 1} \hat{\sigma}_j = \mathsf{tr}(T_n) = \mathsf{tr}(\boldsymbol{K}_n) = \frac{1}{n} \sum_{i=1}^{n} K(X_i, X_i).$$

Compute the trace of the population analogue: for any orthonormal basis $\{\phi_j\}_{j \geq 1}$ of the rkHS $H$

$$\mathsf{tr}(T_H) = \sum_{j \geq 1} \langle T_H \phi_j, \phi_j \rangle_H$$

$$= \sum_{j \geq 1} \left\langle \int_{\mathcal{X}} \phi_j(x) k_x \, \mathrm{d}P(x), \phi_j \right\rangle_H$$

$$= \sum_{j \geq 1} \int_{\mathcal{X}} \phi_j(x) \langle k_x, \phi_j \rangle_H \, \mathrm{d}P(x)$$

$$= \sum_{j \geq 1} \int_{\mathcal{X}} \phi_j(x) \phi_j(x) \, \mathrm{d}P(x)$$

$$= \int_{\mathcal{X}} \sum_{j \geq 1} \phi_j(x) \phi_j(x) \, \mathrm{d}P(x)$$

$$= \int_{\mathcal{X}} K(x, x) \, \mathrm{d}P(x)$$

where we interchanged the integral $\int \, dP$ with the inner product by Bochner integrability, applied the reproducing property of the kernel $k_x$, interchanged the sum with the integral by Fubini's, and used the standard Mercer expansion of the kernel that has uniform convergence.

Define the centered random variables $\zeta_i = K(X_i, X_i) - \mathsf{E}\, K(X, X)$ supported on the interval $[\kappa, \kappa]$, and apply the standard Hoeffding inequality [Wai19, eq2.11]:

$$\left| \sum_{j \geq 1} \hat{\sigma}_j - \sigma_j \right| = \left| \mathsf{tr}(T_n) - \mathsf{tr}(T_H) \right| = \left| \frac{1}{n} \sum_{i=1}^{n} \zeta_i \right| \leq \varepsilon$$

with probability at least $1 - 2e^{-2n\varepsilon^2/(2\kappa)^2}$.

## C.3   Proof of Lemma 4.6

From [RBD10, prop6], for compact positive operators $A, B$

$$\text{if } \|A - B\|_{\mathsf{op}} \leq [\alpha_N - \alpha_{N+1}]/4, \quad \text{then } \|P_D^B - P_N^A\|_{\mathsf{op}} \leq \frac{2}{\alpha_N - \alpha_{N+1}} \|A - B\|_{\mathsf{op}} \quad \text{(C.3)}$$

where $\alpha_N$ and $\alpha_{N+1}$ are the $N$th and $(N+1)$st distinct eigenvalues and $P_N^A$ is the projection on the eigenspace of the top $N$ distinct eigenvalues of $A$, whereas $P_D^B$ is the projection on the eigenspace of top eigenvalues of $B$ of the same dimension. If, in addition, $A, B$ are Hilbert-Schmidt,

$$\text{if } \|A - B\|_{\mathsf{HS}} \leq [\alpha_N - \alpha_{N+1}]/4, \quad \text{then } \|P_D^B - P_N^A\|_{\mathsf{HS}} \leq \frac{2}{\alpha_N - \alpha_{N+1}} \|A - B\|_{\mathsf{HS}}. \quad \text{(C.4)}$$

As [RBD10, thm12] point out, a bound on the projection onto the eigenspace of a simple (multiplicity 1) eigenvalue implies a bound on the eigenfunctions: let $\hat{\phi}, \phi$ be unit-norm and $\langle \hat{\phi}, \phi \rangle > 0$, then

$$\|\hat{\phi} - \phi\|_H^2 = 2(1 - \langle \hat{\phi}, \phi \rangle_H) \leq 2(1 - \langle \hat{\phi}, \phi \rangle_H^2) = \|P_{\hat{\phi}} - P_\phi\|_{\mathsf{HS}}^2.$$

If $2\kappa\sqrt{2\tau}/\sqrt{n} \leq [\sigma_N + \sigma_{N+1}]/4$, then by (C.2) $\|T_n - T_H\|_{\mathsf{HS}} \leq [\sigma_N + \sigma_{N+1}]/4$ with probability at least $1 - 2e^{-\tau}$, and therefore by (C.4)

$$\|P_{\hat{H}_N} - P_N\|_{\mathsf{HS}}^2 \leq \frac{2^2}{[\sigma_N - \sigma_{N+1}]^2} \|T_n - T_H\|_{\mathsf{HS}}^2 \leq \frac{(2\kappa)^2 2\tau}{n} \frac{2^2}{[\sigma_N - \sigma_{N+1}]^2}.$$

This event occurs if $n \geq (2\kappa)^2 2\tau(4)^2/[\sigma_N - \sigma_{N+1}]^2$.

Next, we work with the population orthonormal basis $\{\phi_j := \sqrt{\sigma_j} e_j\}_{j=1}^\infty$ for $H$ and extend the population counterpart $\{\hat{\phi}_j := \sqrt{\hat{\sigma}_j} \hat{e}_j\}_{j=1}^n$ to an orthonormal basis for $H$. This is possible because there are $n$ independent eigenvectors $P$-a.s. by our assumptions that $P$ is continuous and $\boldsymbol{K}_n$ is strictly positive definite.

Using Parseval's identity, and then Parseval's again with the projection operators $(I_H - P_N)$ and $P_N$:

$$\|P_{\hat{H}_N} - P_N\|_{\mathsf{HS}}^2 = \sum_i \left\|(P_{\hat{H}_N} - P_N)\phi_i\right\|_H^2 = \sum_i \left[ \sum_j \left|\langle (P_{\hat{H}_N} - P_N)\phi_i \,,\, \hat{\phi}_j \rangle_H\right|^2\right]$$

$$= \sum_{i,j=1}^{r(N)} 0 + \sum_{i \geq r(N)+1} \left[ \sum_{j=1}^{r(N)} \left|\langle \phi_i \,,\, \hat{\phi}_j \rangle_H\right|^2\right]$$

$$+ \sum_{i=1}^{r(N)+1} \left[ \sum_{j \geq r(N)+1} \left|\langle -\phi_i \,,\, \hat{\phi}_j \rangle_H\right|^2\right] + \sum_{i,j \geq r(N)+1} 0$$

$$= \sum_{j=1}^{r(N)} \left[ \sum_{i \geq r(N)+1} \left|\langle \phi_i \,,\, \hat{\phi}_j \rangle_H\right|^2\right] + \sum_{j \geq r(N)+1} \left[ \sum_{i=1}^{r(N)+1} \left|\langle \phi_i \,,\, \hat{\phi}_j \rangle_H\right|^2\right]$$

$$= \sum_{j=1}^{r(N)} \left[ \sum_i \left|\langle (I - P_N)[\phi_i] \,,\, \hat{\phi}_j \rangle_H\right|^2\right] + \sum_{j \geq r(N)+1} \left[ \sum_i \left|\langle P_N[\phi_i] \,,\, \hat{\phi}_j \rangle_H\right|^2\right]$$

$$= \sum_{j=1}^{r(N)} \left[ \sum_i \left|\langle \phi_i \,,\, (I - P_N)[\hat{\phi}_j] \rangle_H\right|^2\right] + \sum_{j \geq r(N)+1} \left[ \sum_i \left|\langle \phi_i \,,\, P_N[\hat{\phi}_j] \rangle_H\right|^2\right]$$

$$= \sum_{j=1}^{r(N)} \left\|(I - P_N)[\hat{\phi}_j]\right\|_H^2 + \sum_{j \geq r(N)+1} \left\|P_N[\hat{\phi}_j]\right\|_H^2$$

$$\geq \sum_{j=1}^{r(N)} \left\|(I - P_N)[\hat{\phi}_j]\right\|_H^2 + \sum_{j=r(N)+1}^{n} \left\|P_N[\hat{\phi}_j]\right\|_H^2$$

Note that the bound we obtain a bound in terms of the rkHs norm, which implies a counterpart bound for the $L^2(P)$ norm.

## C.4  Proof of Theorem 4.7

Fix $r \geq 1$ and $\lambda \geq 0$, for $j = 1, \ldots, r$, the $\hat{\sigma}_j \xrightarrow{P} \sigma_j$ by Lemma 4.5. Assuming for simplicity that the eigenvalues are distinct, $\|\hat{e}_j - e_j\|_H \xrightarrow{P} 0$ by Lemma 4.6. Recall that also $\hat{e}_j \to e_j$ uniformly on the compact set $\mathcal{X}$. Assuming $f, \hat{f}$ are continuous and $\hat{f} \to f$ $P$-a.s. and $\hat{f}/f$ is bounded on $\mathsf{spt}(f)$, assuming that $\psi_{\hat{f}} \to \psi_f$ in $L^1(f)$; then by dominated convergence

$$\langle \psi_{\hat{f}} \,,\, \hat{e}_j \rangle_{L^2(\hat{f})} = \int_{\mathcal{X}} \psi_{\hat{f}} \hat{e}_j \hat{f} \, \mathrm{d}\mathscr{L}^d = \int_{\mathcal{X}} \psi_{\hat{f}} \hat{e}_j \hat{f}/f \, \mathrm{d}P$$

$$\xrightarrow{P} \int_{\mathcal{X}} \psi_f e_j \, \mathrm{d}P = \langle \psi_f \,,\, e_j \rangle_{L^2(f)}.$$

Conclude that $\|\hat{\psi}_\lambda^r - \psi_\lambda^r\|_H \to 0$ in $P$. For $r_n \to \infty$ slow enough, also have $\|\hat{\psi}_\lambda^{r(n)} - \psi_\lambda^{r(n)}\|_H \to 0$ in $P$. Finally, by the universality of $H$, there exists a sequence $\lambda_n \to 0$ slowly enough such that $\|\hat{\psi}_{\lambda(n)}^{r(n)} - \psi\|_{L^2(P)} \to 0$ in $P$.

## C.5  Toy Monte Carlo experiment

We check our theoretical results with a simple numerical experiment. Let $\theta(P) = \mathsf{E}_P[X]$ be the mean functional. Then $\psi_P(x) = x - \theta(P)$. We use the Gaussian PSD kernel from our Example 4.3 and set the shape parameter $\epsilon = 1$. We simulate Monte Carlo data from the standard Normal distribution, corresponding to the shape parameter $\alpha = 1/\sqrt{2}$ of our Example 4.3. This allows us to compute the oracle $\psi_\lambda^r$ using Hermit polynomials that we numerically evaluate using the MATLAB code provided with the textbook [FM15]. We estimate the eigenvalues $\sigma_j$ and eigenfunctions $e_j(X_i)$

the using Nyström method via MATLAB's `eig` function. We estimate the pathwise derivatives as $\frac{1}{n}\sum_{i=1}^{n} X_i \hat{e}_j(X_i)$, note this does not take into account estimation of the density and evaluation of the mean functional on the estimated distribution.

**Simulation experiments**  As a toy experiment, we compute the oracle low-rank regularization $\psi_\lambda^r$ and its estimator $\hat{\psi}_\lambda^r$ as well as the distribution of the estimation error $\|\psi_\lambda^r - \hat{\psi}_\lambda^r\|_{L^2(P)}$ for the mean functional $\theta = E[X]$ in the setting of Example 4.3.

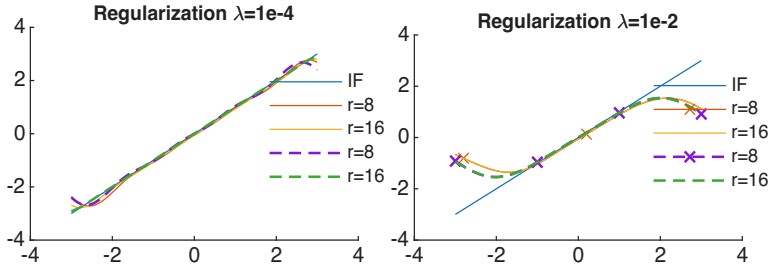

Figure 2: Influence function $\psi$, regularized oracle surrogate $\psi_\lambda^r$ (dashed) and estimate $\hat{\psi}_{\lambda,n}^r$ (solid).

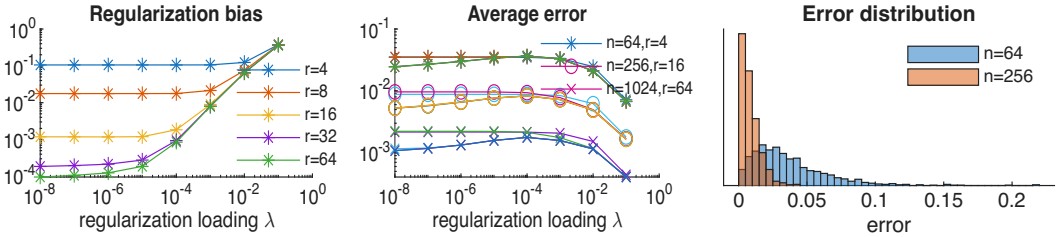

Figure 3:  Surrogate bias $\|\psi - \psi_\lambda^r\|_{L^2(P)}^2$, mean integrated squared error $E\|\psi_\lambda^r - \hat{\psi}_\lambda^r\|_{L^2(P)}^2$, and distribution of the error $\|\psi_\lambda^r - \hat{\psi}_\lambda^r\|_{L^2(P)}$ based on $10^3$ Monte Carlo experiments.

In these experiments we focus on estimating the spectral basis, leaving the development of numerical pathwise derivatives and their estimates to future work. The setting of Example 4.3 allows working with the exact surrogate. We use Riesz' theorem to compute pathwise derivatives as $D\theta[g] = \int \psi \cdot g \mathrm{d}P$ either with Monte Carlo or numerical integration, which is considerably more precise. Figure C.5 shows the bias-variance trade-off of regularization via the loading $\lambda$, and the asymptotic concentration of the distribution of the estimator around the oracle surrogate.

