# OpenReview forum: "Kernel von Mises Formula of the Influence Function"
_NeurIPS.cc/2025/Conference — NeurIPS 2025 poster_

### Official Review · Reviewer_ExzA · 2025-06-13

**Clarity:** 2
**Significance:** 2
**Originality:** 3
**Rating:** 5
**Confidence:** 3

**Summary:**

The paper considers a way to compute the first variation of a statistical estimator in the Fisher-Rao geometry at some probability measure $P$. To this end, the authors first represent the first variation as steepest descent direction among all (normalized) variations. To make the problem tractable, they introduce a regularization of this variational formulation by some Hilbert space norm. In this case the (regularized) first variation can be expanded into an eigenbasis of the Hilbert space.

In practice, the Hilbert space is chosen as RKHS of a Mercer kernel and the eigenbasis as the principal components of the measure. For this case the authors prove that the computation of the eigenvalues of the kernel matrix still leads to a consistent estimator of the eigenvalues of $P$. As a final result they prove in Thm 3.7 that the combination out of cutting off low eigenvalues and estimating the principal components from data still provides a consistent estimator.

**Questions:**

see strenghts/weaknesses

**Ethical Concerns:**

["NO or VERY MINOR ethics concerns only"]

**Final Justification:**

During the discussion phase, the authors addressed several of my comments. In particular, the authors explained their numerical setup and numerically investigated the bias introduced by the regularization. Some loose ends of the paper remain open (dependency on the kernel, theoretical investigation of the bias introduced by the regularization, applications) or cannot be checked during the discussion phase (that the explanations for a broader audience will be improved).

However, I appreciate the authors efforts in the discussion and I acknowledge that some of the loose ends can be classified as "future work". Presuming that the promised changes will be incorporated in the paper and that the accessibility for a broader audience will be improved (which is in the authors own interest), I raised my rating to 5 (accept).

Regarding the scope, I still think that the paper will be very niche at NeurIPS. But the developed method is based on RKHS theory and could be useful for several applications (e.g. for uncertainty quantification or robustness of statistical estimators) and I would therefore see it within the scope of NeurIPS.

**Limitations:**

yes

**Quality:**

3

**Strengths And Weaknesses:**

## General comment

I guess, the paper will be very niche for NeurIPS, but I would see it within the scope of NeurIPS such that there is not too much of a problem.

## Strengths

- I find the concept of discretizing the influence function in terms of eigenfunctions instead of using a spatial discretization quite nice.

- The authors provide a theoretical foundation of their regularization and approximation method including convergence (consistency) results. The technical tools (regularization with an RKHS norm, spectral decomposition) are well-established and quite likely to work.

## Weaknesses

- A critique which might be quite common for theory papers: It is not clear whether the proposed framework works well in practice. There is only one very small 1D example in the paper. It would be beneficial to implement the method in a more serious setting and compare the error to the point evaluation method mentioned in the introduction (which will be of course more expansive when evaluating more points, but is it more accurate?). How does it scale with the dimension of the probability measures (which often is a problem for kernelizations)? Given the theoretical nature of the paper, I wouldn't weight this critique to much, but given that the paper claims to propose a faster and more practical method than previous papers, at least some evaluations could be included.

- I think that the authors could invest a bit more efforts in writing and explaining their results. While the technical part of the paper is well readible (given a general background in analysis), the explanations in the introduction and limitations/future work are very sparse and I am not fully clear what the authors mean even after reading the paper.

- Similarly, the paper would benefit from a how-to section in the paper (potentially in the appendix): That is, a section which says, given data points $x_1,...,x_n$, what do I have to do to get the (approximated) influence function.

- The approximation procedure is specific to the choice of $P$. So if I want to compute the first variation of the same estimator for different base distributions this might be restrictive.

- It remains unclear to me how much bias is introduced by the regularization and how this depends on the kernel/RKHS which is chosen.

## Minor

- line 35: please clarify that the tangent space is the tangent space within Fisher-Rao geometry (for other geometries the tangent spaces look different). Also define $L_0^2$ (probably zero-mean $L^2$-functions?).

- it is a bit strange that the differential operator $D\theta_P$ is introduced in line 142-143 while it is already used before. I would suggest to put the definition directly in Section 2.1.

- Section 1: The $\psi$ is often the index P (unless this is intentional, in which case the authors should explain the difference between $\psi$ and $\psi_P$)

- The simulation experiments at the end of the main paper don't have enough explanations. What do we see in the plots? Please add some description. What does MISE stand for? I would suggest the authors to rather shift this part to the appendix and properly explain it instead of presenting it without explanations in the main part.

- It's "principal components" not "principle components" (appears in line 102, 106, 175, 176)...

---

> ### Author Rebuttal · Authors · 2025-07-31
>
> We thank the Reviewer for the careful reading and insightful interpretation of
> our work, valuable and encouraging feedback.
> In particular, we are delighted that the Reviewer found our work within scope
> for NeurIPS,
> our core idea of discretizing along a basis of eigenfunctions quite nice,
> and our method theoretically well founded,
> the technical part our paper well readable,
> and numerical feasibility of our method quite likely to work.
>
> ### Computational feasibility and comparison to the baseline
>
> We acknowledge that expanding our simulation experiments to include more
> interesting and more complex estimation targets would greatly aid the impact of
> the paper.
> We have experimented with the following more interesting functionals:
> the average density functional from Bickel'88,
> $\theta(P) = E[p(X)^2]$ with $\psi_P(x) = 2p(x) - 2\theta(P)$, where $p$ is the
> density of $P$;
> the median/quantile functional whose influence function contains a jump
> discontinuity;
> the entropy and mutual information functionals (a 2d example);
> the Gini coefficient and Lorenz value B functionals.
> The simulations results are qualitatively the same for these functionals as
> those for the mean, we obtain visually good estimates with very few spectral
> terms, with the exception of the median whose Fourier series converges slower
> due to the discontinuity.
>
> Regarding the computational complexity,
> the cost of our method is incurred with
> density estimation (complexity and accuracy vary depending on the method),
> kernel PCA,
> and evaluation of the functional on the perturbed density, say $C$.
> If we use kernel methods for density estimation, and use Nystr\"om
> approximations for both the density and kPCA estimation, then the
> computational complexity is $O(nr^2 + rC)$.
>
> Regarding comparison to the point-wise baseline method, it too incurs the cost
> of estimating the density, which is at best comparable to the cost of kPCA, so
> it would scale as $O(nr^2 + nC)$ with the same numerical accelerations.
> Please also see our response to Reviewer em1K.
>
> We agree with the Reviewer that the most convincing demonstration of the
> feasibility would be to apply our method to a real-world-scale problem and think
> that this would be a great goal for follow-up work.
> As for the present theoretical paper, we thank the Reviewer for acknowledging
> that kernel methods are well-established and numerical methods that scale them
> are extensively and actively researched.
> Indeed, we are excited to explore applications of [Che+25] and other randomized
> numerical linear algebra methods we recently learned about at the Householder
> Symposium to implementations of our estimator on such problems.
> Please also see our response to Reviwer em1K describing additional experimental
> work we will focus on for the camera-ready version to demonstrate applicability
> to relevant statistical settings.
>
> ### Improving the exposition
> Per your suggestion (and similar feedback of other reviewers) we will
> expand the introduction for the camera-ready version to make the background and
> connection with prior and related works more accessible, invest significant
> effort on explaining our results, clarify our simulations and limitations/future
> work.
>
> ### How-to section
> We will include detailed steps to implement our method in he Appendix for he
> camera-ready version.
>
>
> ### Dependence on $P$
> This is a very keen observation, but we view it as follows.
> The influence function $\psi_P$ that we want to estimate depends on $P$,
> therefore, we see the dependence on $P$ of our eigenbasis as a strength rather
> than a weakness, as it opens the door for adaptation.
> Also, we cannot really think of a setting in statistics where one would want to
> estimate the exact same functional on many different populations, but it's
> possible that such settings exists in other fields.
>
>
> ### Regularization bias and optimal choice of $\lambda$ and kernel
> This is a great question.
> We do not claim to offer any insight into the choice of the regularization
> parameter $\lambda$ of our estimator in this paper.
> The optimal choice would strike a balance between the bias (increases in
> $\lambda$) and the variance (decreases in $\lambda$) and would depend on the
> smoothness of the target IF/functional and $P$.
> A finer analysis of the rate of convergence of our estimator is required for
> this analysis.
> This is a goal of our follow-up work, but requires a lot more theory and
> reasonably out of scope for the present paper as acknowledged by Reviewer em1K.
>
> ### Minor points
> line 35: the tangent space is indeed the one of the Fisher-Rao geometry
> containing all zero-mean $L^2$ functions. We will clarify in the paper.
>
> line 142: our intention was to introduce $D\theta_P$ in the outline of the proof to
> Theorem 1.1, clearly this didn't work, so we will revisit and re-evaluate how we
> can be more effective at introducing this and related background
> notation/concepts in Section 1.
> Any further feedback would be much appreciated.
>
> Plots: per your suggestion and similar feedback from other reviewers we will
> rework and expand the simulations section, improve the readability of the plots,
> and provide an more detailed explanation of the experiments and results in the
> camera-ready version.
> MISE stands for Mean Integrated Squared Error, please also see the response to
> Reviewer 8gPt.
>
> Thank you very much for pointing out the typos in PCA.

---

> > ### Comment · Reviewer_ExzA · 2025-08-01
> >
> > Many thanks for the detailed replies.
> >
> > ### Regarding the dependence on $P$:
> >
> > I guess it will depend very much on the application whether this is a restriction or not. Assuming that there is a fixed dataset and that we want to measure sensitivities of the estimator with respect to certain data points, this might not be too limiting. But if we want to optimize the process generating the data using the functional derivative, this involves to compute the eigenbasis in each step (and therefore I guess that the proposed estimation procedure will become intractable). So, while I acknowledge that there will be applications where this is not an issue, it will be a restriction and I disagree with viewing it as a strength.
> >
> > ### Regarding $\lambda$:
> >
> > To which variance are you referring? The variance introduced by only knowing a finite dataset of samples from $P$ instead of $P$ itself?
> >
> > While I could accept that a theoretical analysis of this bias is hard to access, there are several ways to access my questions in a "lighter way". A detailed discussion that the choice of $\lambda$ is a bias-variance trade-off could be a step in the right direction. Another possibility would be to explore the introduced error (maybe decomposed into bias and variance) in a numerical example.

---

> > > ### Author Response · Authors · 2025-08-04
> > > **Dependence of the basis on $P$ and choice of $\lambda$**
> > >
> > > Excellent point on $P$-dependence. Indeed, for applications to robustness cited on line 25 of the paper, one might use the IF as the gradient direction in a greedy optimization procedure and incur the cost of eigendecomposition (3.4) at each step. Modifying our estimator to perform all/many estimations of the IF using one estimate of the basis would be a useful extension for this setting and should not be hard. For statistical applications of the IF cited on lines 24, 27, 28, only one estimate of the IF is required and this is the setting we had in mind in our response. Many thanks for the comment.
> > >
> > > Regarding the choice of $\lambda$,
> > > by `bias` we mean the error made by replacing the estimation target with the
> > > regularized surrogate $||\psi - \psi_{\lambda}||^2_{L^2(P)}$.
> > > From equation (2.5) in the paper, this is
> > > $$
> > > bias=
> > > C + \sum_{j=1}^r [1-1/(1+2\lambda/\sigma_j)]^2 [d\theta(P^j_t)/dt]^2=
> > > C + \sum_{j=1}^r [1-1/(1+2\lambda/\sigma_j)]^2 C_j
> > > $$
> > > which increases in $\lambda$.
> > >
> > >
> > > By `variance` we mean the Mean Integrated Squared Error (MISE)
> > > $
> > > E ||\hat\psi_\lambda - \psi_\lambda ||^2_{L^2(P)}.
> > > $
> > > If we make the (inaccurate, but very convenient) simplifying assumptions that
> > > (i) the estimates of the $j$th eigenfunction, pathwise derivative, and
> > > eigenvalue are independent across $j=1,\ldots,r$,
> > > and
> > > (ii) the estimator $\hat\psi_\lambda$ is unbiased,
> > > (iii) we can replace $\hat\sigma_j + 2\lambda = \sigma_j + 2\lambda$
> > > then
> > > $$
> > > MISE\approx\sum_{j=1}^r [1/(\sigma_j+2\lambda)]^2
> > > \int Var(
> > >   \hat\sigma_j [d\theta(\hat P^j_t)/dt] \hat e_j(x)
> > > ) dP(x)=
> > > \sum_{j=1}^r [1/(\sigma_j+2\lambda)]^2 C'_j
> > > $$
> > > which decreases in $\lambda$.
> > >
> > > The expected total estimation error $E||\hat\psi_\lambda - \psi||^2_{L^2(P)}$
> > > can be bounded by the sum of these two terms, so the optimal choice of $\lambda$
> > > would balance these two terms.
> > >
> > > Given a smoothness assumption on $\psi$, these terms can be estimated
> > > theoretically in terms of rates, leading to a theoretical rate for optimal $\lambda$.
> > > In practice, one would seek an empirical (based on data) estimate of the total
> > > estimation error and use sample-splitting to choose $\lambda$ based on the data.
> > > However, it is not immediately clear how to form a data-driven estimate of the
> > > estimation error here.
> > >
> > > Below are calculations of `bias` and `MISE` in our experiments.
> > > We will include a discussion of $\lambda$ and calculations of the bias and MISE
> > > in our numerical experiments for the camera ready version of the paper.
> > >
> > > ### Bias
> > > |   | r=4 | 8 | 16 | 32 | 64 |
> > > |---|---|---|---|---|---|
> > > | $\lambda$=5 | 0.96455 | 0.9643 | 0.9643 | 0.9643 | 0.9643 |
> > > | 1 | 0.84801 | 0.84675 | 0.84674 | 0.84674 | 0.84674 |
> > > | 0.5 | 0.73723 | 0.73473 | 0.73472 | 0.73472 | 0.73472 |
> > > | 0.1 | 0.37547 | 0.36401 | 0.36395 | 0.36395 | 0.36395 |
> > > | 0.05 | 0.25104 | 0.23025 | 0.23013 | 0.23013 | 0.23013 |
> > > | 0.01 | 0.12366 | 0.064951 | 0.064383 | 0.064383 | 0.064383 |
> > > | 0.005 | 0.11138 | 0.036246 | 0.03518 | 0.03518 | 0.03518 |
> > > | 0.001 | 0.10606 | 0.011443 | 0.0079241 | 0.0079237 | 0.0079237 |
> > >
> > >
> > >
> > > ### MISE n=64
> > > |   | r=4 | 8 | 16 | 32 | 64 |
> > > |---|---|---|---|---|---|
> > > | $\lambda$=5 | 1.7128e-05 | 1.6776e-05 | 1.6776e-05 | 1.6776e-05 | 1.6776e-05 |
> > > | 1 | 0.00033935 | 0.00033105 | 0.00033107 | 0.00033107 | 0.00033107 |
> > > | 0.5 | 0.0010531 | 0.001022 | 0.0010221 | 0.0010221 | 0.0010221 |
> > > | 0.1 | 0.0071162 | 0.0065873 | 0.0065884 | 0.0065884 | 0.0065884 |
> > > | 0.05 | 0.011488 | 0.010028 | 0.010031 | 0.010031 | 0.010031 |
> > > | 0.01 | 0.021789 | 0.014867 | 0.014873 | 0.014873 | 0.014873 |
> > > | 0.005 | 0.025216 | 0.015598 | 0.015564 | 0.015565 | 0.015565 |
> > > | 0.001 | 0.02927 | 0.016356 | 0.015773 | 0.015773 | 0.015773 |
> > >
> > >
> > >
> > > ### MISE n=256
> > > |   | r=4 | 8 | 16 | 32 | 64 |
> > > |---|---|---|---|---|---|
> > > | $\lambda$=5 | 4.5724e-06 | 4.4925e-06 | 4.4924e-06 | 4.4924e-06 | 4.4924e-06 |
> > > | 1 | 9.0695e-05 | 8.8813e-05 | 8.8809e-05 | 8.8809e-05 | 8.8809e-05 |
> > > | 0.5 | 0.00028185 | 0.00027477 | 0.00027476 | 0.00027476 | 0.00027476 |
> > > | 0.1 | 0.0019279 | 0.0018051 | 0.0018047 | 0.0018047 | 0.0018047 |
> > > | 0.05 | 0.0031522 | 0.0028077 | 0.002806 | 0.002806 | 0.002806 |
> > > | 0.01 | 0.0061521 | 0.0044261 | 0.0043875 | 0.0043875 | 0.0043875 |
> > > | 0.005 | 0.0071526 | 0.0046951 | 0.0045757 | 0.0045757 | 0.0045757 |
> > > | 0.001 | 0.0083208 | 0.0051297 | 0.0043872 | 0.0043872 | 0.0043872 |
> > >
> > >
> > >
> > > ### MISE n=1024
> > > |   | r=4 | 8 | 16 | 32 | 64 |
> > > |---|---|---|---|---|---|
> > > | $\lambda$=5 | 1.1517e-06 | 1.1305e-06 | 1.1304e-06 | 1.1304e-06 | 1.1304e-06 |
> > > | 1 | 2.289e-05 | 2.2391e-05 | 2.2388e-05 | 2.2388e-05 | 2.2388e-05 |
> > > | 0.5 | 7.1314e-05 | 6.9446e-05 | 6.9435e-05 | 6.9435e-05 | 6.9435e-05 |
> > > | 0.1 | 0.00049595 | 0.00046446 | 0.00046423 | 0.00046423 | 0.00046423 |
> > > | 0.05 | 0.0008205 | 0.00073437 | 0.00073354 | 0.00073354 | 0.00073354 |
> > > | 0.01 | 0.0016309 | 0.00124 | 0.0012238 | 0.0012238 | 0.0012238 |
> > > | 0.005 | 0.0018996 | 0.0013728 | 0.0013241 | 0.0013242 | 0.0013242 |
> > > | 0.001 | 0.0022108 | 0.0016146 | 0.0013234 | 0.0013235 | 0.0013235 |

---

> > > > ### Comment · Reviewer_ExzA · 2025-08-05
> > > >
> > > > Thank you very much for the discussion and detailed explanations.
> > > >
> > > > Presuming that the promised changes will be incorporated in the paper and that the accessibility for a broader audience will be improved, I raise my score by one point.

---

> > > > > ### Author Response · Authors · 2025-08-08
> > > > >
> > > > > We want to thank the reviewer for carefully reading the paper, providing
> > > > > insightful feedback and making thoughtful suggestions for improving the
> > > > > exposition which will be incorporated in the revision of the paper.

---

### Official Review · Reviewer_em1K · 2025-06-16

**Clarity:** 3
**Significance:** 2
**Originality:** 3
**Rating:** 4
**Confidence:** 4

**Summary:**

The paper proposes a novel way of deriving the influence function of a target parameter via the Riesz representer of its derivative operator. Previous results had focused on point-wise derivations of the influence function (von Mises formula, Theorem 1.1 in the paper), which allow for point-wise approximations of the influence function. In contrast, the ideas presented in this contribution open the door to simultaneous  approximations to all values of the influence function, given that the approximation is conducted via the influence function spectral representation.

The overall idea builds on an approximated variational representation of the influence function: the influence function can be exactly represented as a functional optimization program; this program can be further regularized with a Sobolev norm to obtain approximate solutions. In order to do so, the spectral representation of the regularized functional is exploited. In particular, if this Sobolev space is a Mercer RKHS, then the solutions are actionable in practice. The consistency of the kernel von Mises estimator is established.

**Questions:**

My questions have been posed as the two main weaknesses above. I believe they are concise. If they are addressed satisfactorily, I will raise my score.

**Ethical Concerns:**

["NO or VERY MINOR ethics concerns only"]

**Final Justification:**

The paper is well written overall and technically sound. The main contribution seems original and powerful enough for the paper to be accepted. However,  and as explained in my review, I think the contributions of this paper are not extremely significant until proven practically useful. Based on the rebuttal, I trust that the authors have run further experiments, but it is hard to judge without seeing the actual results. Thus, I believe a borderline accept is a sensible score on my end.

**Limitations:**

The authors have adequately addressed the limitations of their work.

**Paper Formatting Concerns:**

No formatting concerns.

**Quality:**

3

**Strengths And Weaknesses:**

**Strengths**: The paper is well written overall, reading very well despite being technical in nature. The submission is technically sound, with all the claims being theoretically supported. In this sense, the authors also clearly state the main technical/theoretical limitation of the work: they only prove consistency of the estimator, without studying the convergence rates of it. However, the idea seems original and powerful enough so that consistency of the estimator suffices for a first paper on this idea.  While going from point-wise evaluations of functions to spectral approximations of such functions is not extremely original by itself, the combination of these ideas with the variational representation of the influence function makes the contribution non-trivial and original.

**Weaknesses**: While the originality of this paper is not under question, I think these spectral approximations are not substantially significant until proven practically useful. Based on this comment, I see two main flaws in the current version of the paper.

- First, I would like to see a discussion on how easy/computationally feasible it is to obtain the pathwise derivatives $D\theta_P$ on the eigenfunctions $\hat e_j$ in Section 3. Do these need to be numerically approximated themselves? Overall, what is the complexity of the whole procedure?
- Second, the simulations are only run with $\theta$ taken to be the mean functional, arguably the easiest possible functional. I would like to see the performance of these approximations in more challenging/real scenarios. For example, given the importance of influence functions in causal inference and their connection to doubly robust estimators, the authors could take $\theta$ to be the average treatment effect of some causal experiment, and compare the results using the approximate influence function of this paper with the usual plug-in  or inverse probability weighting estimators (as if the doubly robust estimator was not known). I would also encourage the authors to run a simulation on a causal target whose influence function is actually unknown, and compare the results obtained with the approximate influence function obtained using the techniques presented in this paper with some naive estimator of such a causal target.

**Minor comments**:
- Discuss the choice of $r(N)$, both theoretically (does Lemma 3.6 hold for any definition of $r(N)$?) and practically (are practitioners encouraged to used r(N)-rank approximations in practice?).
- It almost seems like the authors run out of space. Theorem 3.7 is hanging there without any context or explanation. Please make some space for explaining what Theorem 3.7 is doing there and its connection to previous results (I do understand it but it does not read well).
- The definition of $D\theta$ first appears in Section 2.3, but the concept is already used in Equation (2.2). Please make the definition appear before it is used.
- The plots are too small, especially the legends.

---

> ### Author Rebuttal · Authors · 2025-07-31
>
> We thank the Reviewer for the careful reading and insightful interpretation of
> our work, valuable and encouraging feedback.
> In particular, we are delighted that the Reviewer found our paper well-written,
> and our variational characterization, its spectral solution and
> its implementation with rkHs projections to be original and nontrivial results.
>
> ### Computational feasibility
>
> As with the baseline method of [CLV19, JWZ22] that are based on the von Mises
> formula (Theorem 1.1),
> our spectral estimator of the IF requires evaluation and estimation of pathwise
> derivatives $d\theta(P_t)/dt$.
> We claim no direct contribution to this subtask.
> Prior works use the finite difference
> $[\theta(\hat f^{\delta,z}_t) - \theta(\hat f)]/t$
> with
> $f^{\delta,z}_t(x) = (1-t)f(x) +t K^{\delta,z}(x)$
> from line 60 in the paper being a mixture perturbation of the density $f$
> toward the mollified point mass at $z$.
> We follow the same basic strategy and approximate $d\theta(P_t)/dt$ with a
> finite difference
> $[\theta(\hat f^j_t) - \theta(\hat f)]/t$
> where
> $\hat f^j_t = [1+t\hat e_j]\hat f$
> is the perturbation toward the $j$th eigenfunction.
> An indirect advantage of our approach to this basic strategy, is that the scores
> $e_j$ for our differences are well-behaved rkHs functions (e.g. Hermite
> polynomials) and our differences are numerically stable.
> By contrast, the former approach requires a careful tuning of the mollification
> parameter $\delta$ and leads to finite differences that are ill-conditioned [JWZ22,
> direct communication with the authors].
>
> So, the cost here is that of density estimation (complexity and accuracy vary
> depending on the method), kernel PCA, and evaluation of the functional on the
> perturbed density, say $C$.
> If we use kernel methods for density estimation, and use Nystr\"om
> approximations for both the density and kPCA estimation, then the
> computational complexity is $O(nr^2 + rC)$.
>
> A better approach would be a composable numerical implementation of the
> functional evaluation,
> the density estimate,
> and the kernel PCA that would allow automatic differentiation with respect to
> $t$ instead of finite differences.
> It is out of scope for this paper but would be an interesting topic for
> follow-up work.
>
> ### More challenging/real scenarios
>
> We acknowledge that expanding our simulation experiments to include more
> interesting estimation targets would greatly aid the impact of the paper.
> We have experimented with the following more interesting functionals:
> the average density functional from Bickel'88,
> $\theta(P) = E[p(X)^2]$ with $\psi_P(x) = 2p(x) - 2\theta(P)$, where $p$ is the
> density of $P$;
> the median/quantile functional whose influence function contains a jump discontinuity;
> the entropy and mutual information functionals;
> the Gini coefficient and Lorenz value B functionals.
> The simulations results are qualitatively the same for these functionals as
> those for the mean, we obtain visually good estimates with very few spectral
> terms, with the exception of the median whose Fourier series converges slower
> due to the discontinuity.
>
> We fully agree with the Reviewer that applying our method to estimation of
> average treatment effect (ATE) functionals would be a powerful test of the
> theory and an impactful demonstration of significance of our work.
> A good baseline for such experiment is the Kernel Debiased Plug-in
> Estimator (KDPE) of [Cho+24] which implements the debiased estimators of
> treatment effect functionals without requiring the influence function as input
> and compares them with the well-known Targeted Maximum Likelihood estimator
> (TMLE) that requires the influence function as input.
> The paper considers both a straightforward experimental setup as well as a
> longitudinal design.
> As [JWZ22] and [Cho+24] point out, working out the influence function and an
> efficient estimator in the longitudinal setting is rather complicated, making
> for a convincing case for automated estimation of the IF and debiasing.
>
> For the camera-ready version of the paper we will include experiments that will
> compare the accuracy in downstream tasks of estimating functionals ATE, RR, OR
> from page 8 in [Cho+24]
> using TMLE with the analytically estimated IF, TMLE with the IF estimated using
> our method, and KDPE that does not require the IF.
>
> ### Choice of $r(N)$
> To fully address the choice of the number of basis functions $r(N)$ and the
> choice of regularization parameter $\lambda$, a finer analysis of the rate of
> convergence of the estimator is required.
> One can further consider the trade-off between statistical precision and
> numerical complexity.
> For Lemma 3.6, the finite-sample concentration bound requires that
> $n> 128\kappa^2\tau/[\sigma_N - \sigma_{N+1}]^2$ the sample size be larger
> enough relative to the spectral gap.
> In our simulations we have found that as few as $r=8$ terms are sufficient to
> approximate the overall shape of $\psi_P$ visually and we expect that the
> practical choice $r$ to be relatively low in applications.
>
> ### Context of Theorem 3.7
> We completely agree and will invest significant effort in improving the
> exposition throughout the paper for the camera-ready version.
> In particular, we will expand the explanation, context and interpretation of
> Theorem 3.7.
>
> ### Definition of $D\theta$
> Thank you for pointing this out, our intention was to introduce $D\theta_P$ in
> the outline of the proof to Theorem 1.1, but we realize from your and Reviewer
> ExzA feedback that we didn't succeed.
> We will address this in the camera-ready version.
> Any further feedback would be much appreciated.
>
> ### Clarity of plots
> Per your feedback and similar from other reviewers, we will be expanding the
> experiments section and will include more explanations and update the plots to
> be legible for the camera-ready version.

---

> > ### Comment · Reviewer_em1K · 2025-08-05
> >
> > I thank the authors for the careful reply. My main concern (i.e., applicability of the method) has been addressed. I trust the authors will make the changes for the camera-ready version.  I will update my scores accordingly.

---

> > > ### Author Response · Authors · 2025-08-08
> > >
> > > We want to thank the reviewer for carefully reading the paper, providing
> > > insightful feedback and making thoughtful suggestions for improving the
> > > exposition which will be incorporated in the revision of the paper.

---

### Official Review · Reviewer_8gPt · 2025-06-21

**Clarity:** 2
**Significance:** 3
**Originality:** 3
**Rating:** 3
**Confidence:** 1

**Summary:**

This work studies estimating the influence function of functionals that are asymptotically linear. The work introduces an estimator for the influence function by approximating changes in the influence function from the derivatives of a kernel estimator. The authors prove that their estimator is consistent and provide some experimental demonstration of their approach.

**Questions:**

1.	How strong is this assumption that our estimators are asymptotically linear?
2.	How can the estimators error be represented by contributions of individual data points in a way that is independent of the total number of samples? Is this an assumption as well? If so, how is this assumption motivated?
3.	Can influence functions in sample distribution space be connected somehow to influence in input space? I.e. instead of looking at perturbations in the distribution of the data, what if you examined the influence of perturbing a sample in a particular direction in its ambient space.

**Ethical Concerns:**

["NO or VERY MINOR ethics concerns only"]

**Final Justification:**

I found it difficult to judge the novelty and significance of the contributions in this work. I think the style, content, and format of the work may be a better fit for other more statistically oriented venues and not the broader NeurIPS audience.

**Limitations:**

The biggest limitation of this work is that the majority of the NeurIPS audience will likely have difficulty appreciating this result. I recommend spending significant effort to clarify the background and novelty of their results, as well as possibly expanding the experimental section to include comparisons to previous methods. If previous methods don’t exist or are unimplementable for some valid reason, then an explanation of this fact is sufficient.

**Paper Formatting Concerns:**

The authors should double check the references are formatted according to the NeurIPS guidelines.

**Quality:**

3

**Strengths And Weaknesses:**

I am not familiar with the literature, so I may be incorrect on some of these criticisms and strengths. If the authors can address my confusion, I may be willing to raise my score.

Strengths:

-	The problem of estimating the influence function seems fundamental and well-motivated.

-	Their approach seems novel and theoretically grounded.

-	Some experimental validation of their method is provided.

Weaknesses:
-	I am having difficult placing the novelty of the result in the context of prior work. How has the influence function been estimated in prior works and how does this approach compare?

-	The plots are largely unclear. What are the axes labels? What should the reader take away from these plots? How should the reader interpret the magnitude of the errors in the estimate of the influence function? What about functionals other than the mean?

-	It would be good to provide some explanation of what the ‘Von Mises’ formula is for the reader.

-	It is not clear to me how to compute the derivatives of your functional. This needs to be clarified, perhaps with an example.

---

> ### Author Rebuttal · Authors · 2025-07-31
>
> Thank you for the careful reading and thoughtful feedback, and for
>  acknowledging that our problem is fundamental and well-motivated.
>
> ### Novelty and Relation to Prior Work
> We agree that the Introduction could make the connection to prior work more explicit, and we will revise it accordingly for the camera-ready version. Below we expand on this connection to clarify the novelty and contribution of our work.
>
>
> *What is the influence function (IF) and why is it important?*
>
> It is a classical result that the MLE is efficient—it attains the smallest possible asymptotic variance among regular estimators. In parametric models, this is a simple exercise in calculus: the score function (the derivative of the log-likelihood) has variance equal to the Fisher information, and its inverse provides the lower bound on estimator variance. The score, up to normalization, is the influence function.
>
> However, in nonparametric models, there is no canonical parametrization to carry
> out the analysis, necessitating more advanced theory [Bic+93,Vaa00].
>
> In nonparametric models, however, there is no canonical parametrization, and one must resort to more advanced theory [Bic+93, Vaa00]. The key object in this theory is the IF $\psi_P$​, which quantifies the sensitivity of a functional θ(P) to local perturbations of the distribution P. Its L2-norm gives the lower bound on the variance of any regular estimator and is central to constructing efficient estimators. Deriving $\psi_P$ is a fundamental task, analogous to solving a differential equation—each model requires distinct analytical tools, and obtaining a new influence function is a substantive contribution.
> Closed-form expressions may not even exist.
>
> *How does one estimate the IF?*
>
> In parametric models, this task is straightforward: differentiate the log-likelihood, compute the information, normalize the score, and plug in parameter estimates. Recent work by R. Agrawal et al. (NeurIPS'24) automates this process using autodiff and probabilistic programming.
>
> In nonparametric models, this is much more complex.
> One typically begins by deriving the analytic form of the IF
> using pathwise derivatives along a parametric submodel.
>
> One typically begins by computing the derivative $d\theta(P_t)/dt$
> along a parametric submodel with known score function $g$ and then attempts to express this derivative as
> $E[\phi \cdot g]$ to identify $\psi_P = \phi$ via Riesz representation.
> The form of $\psi_P$ often includes unknown components (e.g.,
> densities, regression functions) that must then be estimated.
>
> *How does this paper relate to prior work?*
>
> Several recent works—[CLV19], [Hin+22], [Ken24], [JWZ22, M Jordan et al
> NeurIPS'22]—note the challenge of deriving IFs and propose
> methods for automating this task.
> These works aim to bypass the functional analytic derivation by replacing it with standard statistical tasks like density estimation.
> However, these methods only yield pointwise evaluations $\psi_P(x)$,
> not functional estimators, and they do not offer convergence guarantees in function norms—guarantees that are crucial for downstream tasks like debiased machine learning (DML).
> By contrast, we develop a novel representation that treats $\psi_P$
> as an element of a Hilbert space and construct a data-driven estimator of the entire function,
> with convergence guarantees in L2 and rkHs norms.
> We are not aware of any prior work that achieves this.
>
> *Contributions of This Paper*
>
> (I) Spectral representation:
> We derive a regularized representation (Theorem 2.3) that generalizes von
> Mises formula (Theorem 1.1) by replacing pointwise perturbations with an
> expansion in a Hilbert space $H$.
> This yields a formula for all values of $\psi_P$ simultaneously, avoiding calculation per point $x$.
> Compared to von Mises, which perturbs the measure $P$ sharply near $x$, our method perturbs $P$ along smooth basis functions,
> which is numerically stable and well-aligned with functional approximation
> theory.
>
> (II) Estimation via rkHs approximation and kPCA:
> We construct a data-driven estimator of $\psi_P$ by
> taking $H$ to be an rkHs, and estimating the basis functions
> via kernel PCA—a well-studied (in NeurIPS) data analysis task.
> This yields a fully data-driven functional estimator of the IF.
> We prove consistency (Theorem 3.7), and argue that this
> approach is practical, scalable, and grounded in rkHs theory and numerical
> analysis.
>
> (III) This combination of our spectral representation with kernel PCA enables the use
> of mature and rich theoretical and numerical tools.
> For theory, it allows the use of results on operator concentration.
> For computation, it allows the use of scalable kernel methods (low-rank
> approximations, randomized numerical linear algebra, fast kernel matrix
> computation).
>
>
> While we focuses on consistency, we also lay the groundwork for
> further analysis of convergence rates, Gaussian approximations, scalable computation, and downstream applications.
> These are all directions for follow-up work.
>
> To our knowledge, no existing method provides a data-driven functional estimator
> of the IF or convergence guarantees in rkHs norms,
> nor has it been shown how to use kernels to estimate the IF.
>
> As Reviewers em1K and ExzA note, our approach rethinks the discretization of the IF—shifting the paradigm.
>
> ### Plots, briefly
>
> We will expand the section on experiments in the camera-ready version accordingly.
> In the following, we provide some more details.
>
> Top row:
> the IF (straight line), the regularized surrogate (smooth lines)
> and estimates (jagged lines).
>
> The estimates are consistent for the surrogates rather than the IF and the
> degree of regularization (via $\lambda$) controls bias (less/more on the left/right
> plots).
>
> The estimates based on 8-16 basis functions and are
> visually indistinguishable from targets.
> The takeaway is that very few basis functions can be sufficient to form a good
> estimate.
>
> Bottom row: the distribution of the error $||\hat\psi-\psi||$
> for sample sizes of $256$ and $1024$ shows that with more data, the mean
> integrated squared error (MISE) and the distribution of error both shift toward 0,
> as guaranteed by our result.
> On the x-axis is the value of the error and on the y-axis is the frequency of
> that value.
>
> In order to judge the magnitude of the error, an efficiency theory for this
> functional estimation problem is required.
> We will investigate the best possible rate of convergence in follow-up work.
>
> Per your suggestion (and that of PMrS),
> we are including an example of a non-trivial IF in Section 1.1
> and the simulations.
> A good choice is the average density functional from Bickel'82,
> $\theta(P) = E[p(X)^2]$ with $\psi_P(x) = 2p(x) - 2\theta(P)$, where $p$ is the
> density of $P$.
> The simulations results are qualitatively the same for this functional (also for
> the median whose IF contains a jump discontinuity, the entropy and mutual
> information, Gini coefficient and Lorenz value B functionals).
>
>
> ### Computation of pathwise derivatives
>
> Regarding computation of derivatives $d\theta(P_t)/dt$,
> we claim no novelty or contribution here, but agree with the reviewer and will
> clarify how this is done in our and prior work.
> The most basic approach is to compute the finite difference
> $[\theta(\hat f_t) - \theta(\hat f)]/t$
> for a small $t$ and density estimates $\hat f$ for the data and
> $\hat f_t = [1+t\hat e_j]\hat f$ for the perturbation toward basis function
> $e_j$.
> A better approach would use automatic differentiation, but this is functional-
> and implementation-specific and outside the scope here.
>
> Regarding providing more explanation of the von Mises formula, we would love to
> improve,
> and kindly ask Reviewer to clarify exactly what would be helpful to explain?
> We outline how the formula works via the proof sketch to Theorem 1.1 and
> spend lines 79-83 interpreting the formula toward our estimation task.
> More input about what is not clear or not covered would help us,
> and we hope to iterate on this in the discussion.
>
> ### Asymptotic linearity
>
> It is an assumption on both the functional $\theta(P)$ and the estimator
> $\hat\theta$,
> but not a restrictive one.
> For $\theta$ to have an IF it must vary smoothly when $P$ is perturbed.
> Functionals that arise through taking averages with respect to $P$ are smooth,
> but the density $\theta(P)=p(x_0)$ at a fixed point is not smooth.
> Consequently, averages are easier to estimate than density values.
> [Vaa91] shows that existence of IF for $\theta(P)$ is equivalent to the existence
> of an estimator $\hat\theta$ that converges at the fast parametric root-n rate.
> So, if it is expected that $\theta$ can be estimated at the roon-n rate, then it is
> expected to have an IF and admit regular estimators that satisfy (1.1).
> Not all estimators of smooth functionals are regular, e.g. shrinkage estimators
> are not.
> One can construct regular estimators of the average of a regression function
> that is estimated with shrinkage as in DML.
>
> Regarding the error contribution of a single $X_i$, it does depend on $n$ in
> (1.1) and is approximately $\psi_P(X_i)/n$.
> E.g., for the sample mean, the error of an observation is $(X_i-\theta)/n$ for
> any $n$, and (1.1) states that for smooth functionals, there exists estimators
> for which a similar representations holds asymptotically.
> Please let me know if further clarification is needed about this.
>
> ### Transport perturbations
> The IF describes the change in $\theta$ when the probability weight of $x$ is
> perturbed.
> There exists a related object, mentioned on line 90, called the Wasserstein
> gradient vector field (WGVF), that describes the change in $\theta$ to perturbations of
> $P$ by transporting the particles of mass continuously in the sample space.
> The WGVF is related to the influence function via
> $\nabla_W \theta(x) = D_x \psi_p(x)$
> and describes the sensitivity of $\theta$ to perturbations of the location of mass
> at $x$ in the sample.
> In fact, there is a one-to-one correspondence between the IF and the W gradient.

---

> > ### Comment · Reviewer_8gPt · 2025-08-05
> > **Reviewer response**
> >
> > Thank you for your thorough reply.
> >
> > Frankly, I agree with Reviewer PMrS, I feel this paper might not be a good fit for NeurIPS based on the content and style alone. I am having difficulty evaluating the contributions of this paper.
> >
> > Currently I am leaning toward maintaining my score, but with low confidence.

---

> > > ### Author Response · Authors · 2025-08-08
> > >
> > > We want to thank the reviewer for carefully reading the paper, providing
> > > insightful feedback and making thoughtful suggestions for improving the
> > > exposition which will be incorporated in the revision of the paper.

---

> ### Author Response · Authors · 2025-08-05
> **Reiterating request for feedback on the exposition**
>
> To re-emphasize the point made in the rebuttal, in terms of the content and scope,
> (i) the paper contributes to the problem studied in [JWZ22, M Jordan,
> NeurIPS'22] and reference [1, Kandasamy, NeurIPS'15] pointed out by Reviewer
> NjaD;
> (ii) IF-based methods have been studied extensively in recent NeurIPS
> publications (search on this key word alone return a dozen papers in
> NeurIPS proceedings and many more papers use similar ideas like the
> transportation perturbations discussed in the review/rebuttal but with different
> terminology);
> (iii) our theoretical contribution relies on kernel PCA, which has been
> extensively developed in NeurIPS, and we contribute to this strand of NeurIPS
> literature as well.
>
> We very much would like to improve the exposition/style of the paper and make it more
> accessible to a wider NeurIPS audience.
> We will expand our introduction and motivation of the problem and the
> explanations of our results as discussed in our rebuttals.
> We kindly ask again for feedback about what is not clear or requires more
> explanation in the paper and in our rebuttal, and about what is helpful in our
> rebuttal and should be included in the camera-ready version of the paper.
> We were very much hoping for more feedback from the Reviewer on this and kindly
> ask for it again. Thank you very much for helping us improve the paper.

---

### Official Review · Reviewer_NjaD · 2025-06-30

**Clarity:** 3
**Significance:** 3
**Originality:** 3
**Rating:** 5
**Confidence:** 2

**Summary:**

This article introduces kernel-based estimators for the influence function, which is well-known in semiparametric and non-parametric statistics. The authors show that their procedure yields asymptotically consistent estimators and they numerically validate their approach on a toy experiment.

**Questions:**

Key points:

- How do we know that the RKHS $\mathcal H$ is dense in the tangent space $L_0^2(P)$? Do you implicitly appeal to Theorem 4.26 in [1] and are the assumptions there satisfied?
- Similarly, in Assumption 3.1: Which definition of universality do the authors use? The usual ones consider denseness in the space of continuous functions.
- Does the normalization for obtaining (ii) given in line 222 preserve/imply (iii)?
- The "extension operators" $R_n$ seem to be the similar to the usual sampling operators; see [2] and [Lemma A.7, 3].

[1] Steinwart, I. & Christmann, A. (2008). Support vector machines. Springer.

[2] Smale, S., & Zhou, D. X. (2007). Learning theory estimates via integral operators and their approximations. _Constructive approximation_, _26_(2), 153-172.

[3] Sterge, N., & Sriperumbudur, B. K. (2022). Statistical optimality and computational efficiency of Nystrom kernel PCA. _Journal of Machine Learning Research_, _23_(337), 1-32.


Further questions/remarks:

1. The relationship to [1] could be made explicit, or, at least, the reference could be included as from a quick gScholar search this seems to be one of the few NeurIPS publications also considering the influence function.
2. Line 21, "foundational role in theory" => add a few relevant references.
3. "Dilated"/"mollified" seems to be used interchangeably.
4. In line 76, $L_\text{loc}^1(\mathbb R^d)$ is not introduced.
5. In the paragraph starting from line 87, what is the difference of $\psi$ and $\psi_P$ (where the latter was used before)?
6. In line 108, "kPCA" not introduced.
7. In the paragraph starting from line 198, the notations for $K$ and $\phi$ clash.
8. Assumption 3.1: absolutely continuous w.r.t.?
9. The discussion from line 266 could come before Lemma 3.4 to motivate the result.

[1] Kandasamy, K., Krishnamurthy, A., Poczos, B., & Wasserman, L. (2015). Nonparametric von Mises estimators for entropies, divergences and mutual informations. _Advances in Neural Information Processing Systems_, _28_.

A few notes on typos/grammar:
- In line 43, add $\hat \cdot$ to $\theta'$ and $\theta''$.
- In line 53, "is an absolutely".
- In line 100, check grammar/formulation.
- In my opinion, "rkHs" should be "RKHS". The former looks weird.
- In line 174, add "RKHS; recalled below in Section 3.1" or something similar.
- The section titles in Section 3 have full stops at the end.
- In line 308, capitalize "consistency".
- In line 312, typo in "Nyström".
- In line 317, typo in "bounded".
- In line 328, "describe" => "described".
- In line 328, typo in "debiased".

**Ethical Concerns:**

["NO or VERY MINOR ethics concerns only"]

**Final Justification:**

As in my original review, I think this is a very well written article, bridging classical statistical theory and reproducing kernel Hilbert space methods commonly employed in machine learning. While I see the concerns of another reviewer w.r.t. the topic being niche at NeurIPS, I also find the article to be a valuable contribution to machine learning theory and NeurIPS has published works on (and methods using) the influence function in the past.

**Limitations:**

Yes.

**Paper Formatting Concerns:**

No concerns.

**Quality:**

4

**Strengths And Weaknesses:**

### Strengths

- Very well written article, bridging classical statistical theory and reproducing kernel Hilbert space methods commonly employed in machine learning.
### Weaknesses

- No finite sample guarantees. As classical theory [1] considers asymptotic results only, are there existing results to obtain rates, as hinted at in line 110?
- The results apply to compact sample spaces only. While the authors hint at possible generalizations [2], their theory does not take these adjustments into account.

[1] Van der Vaart, A. W. (2000). Asymptotic statistics (Vol. 3). Cambridge University Press.

[2] Steinwart, I., & Scovel, C. (2012). Mercer’s theorem on general domains: On the interaction between measures, kernels, and RKHSs. _Constructive Approximation_, _35_, 363-417.

---

> ### Author Rebuttal · Authors · 2025-07-31
>
> We thank the Reviewer for the careful reading and insightful interpretation of
> our work, valuable and encouraging feedback.
> In particular, we are delighted that the Reviewer found our work to be well
> written and connection between classical statistical theory and rkHs methods
> interesting.
> We also thank the Review for graciously catching and reporting our typos.
>
> ### Weaknesses
>
> It would be helpful to clarify which finite sample guarantees the Reviewer is
> asking about?
> It is a basic limitation of all statistical methods that use the influence
> function to be approximate and asymptotic, as we explain in Limitations and
> future work paragraph.
> However, we do provide finite sample concentration bounds for estimation of the
> eigenfunctions and eigenvalues, and in principle this analysis can be extended
> to establish finite sample bounds for our estimator.
> Please let us know if further clarification is necessary.
>
> ### Key points
>
> We require that the rkHs $H$ be dense in $L^2$, it then, indeed, follows readily
> that the normalization (ii) produces a subspace that is dense in $L^2_0$.
> Since continuous functions are dense in $L^2$, any definition of universality
> should work for our estimator, but what we require precisely is that any score
> function can be approximated as stated on line 219.
>
> ### Further questions/remarks
>
> Thank you for pointing out reference [1] in NeurIPS on estimating the influence
> function, this is indeed a great reference to cite in our paper and discuss in
> our introduction.
>
> line 21: we will be expanding the introduction to be more accessible to a wider
> audience per feedback of multiple reviewers.
>
>
> $L^1_{loc}$ denotes the space of locally integrable functions.
>
> $\psi$ refers to the influence function $\psi_P$, we drop the subscript for ease
> of notation, but realize that it is confusing for some of the reviewers and will
> alert the reader to this in the camera-ready version.
>
> line 108: we will replace kPCA with kernel PCA.
>
> line 198: we use $\phi$ as a generic function of $H$ in this paper, so the
> notation should be fine, but we realize that $\phi$ is commonly used to denote
> the canonical features in rkHs references and may be confusing for that reason.
> We will rethink our choice of notation.
>
> Assumption 3.1: absolutely continuous with respect to the Lebesgue measure.
>
> line 266: this is a valid point, we will expand the discussion before Lemma 3.4
> to motivate the result for our problem.

---

> > ### Comment · Reviewer_NjaD · 2025-08-01
> >
> > I thank the authors for their answers and would like to keep my already positive score.

---

> > > ### Author Response · Authors · 2025-08-08
> > >
> > > We want to thank the reviewer for carefully reading the paper, providing
> > > insightful feedback and making thoughtful suggestions for improving the
> > > exposition which will be incorporated in the revision of the paper.

---

### Official Review · Reviewer_PMrS · 2025-07-04

**Clarity:** 2
**Significance:** 3
**Originality:** 2
**Rating:** 4
**Confidence:** 2

**Summary:**

In the work the authors introduce an estimator for the influence function $\psi_P$, for some statistical $\theta(P)$ with statistical estimator $\hat{\theta}(X_1,\ldots, X_n)$ and the influence function is equal to $\sqrt{n}(\hat{\theta} - \theta(P)) = \frac{1}{\sqrt{n}} \phi_P(X_i)$ (according to (1.1) in the paper). The influence function is useful to know for a variety of statistical tasks. The influence function can be evaluated for specific points in a fairly straightforward way, but it is useful to have a full functional representation of it. In this work the authors introduce an estimator for the influence function using a sort of RKDE kernel basis (i.e. it uses a truncated bases) to estimate $\psi$ from data.

**Questions:**

* I’d really like to see a concrete example—a specific dataset and task—where your method truly enables something new to be done, or clearly outperforms previous benchmarks.
* Can you elaborate more on the tasks where the influence function is important? Maybe full, concrete example.

**Ethical Concerns:**

["NO or VERY MINOR ethics concerns only"]

**Final Justification:**

I appreciate the author's new introduction, which I think is helpful. Nonetheless I find the style, topicality, and presentation somewhat problematic for NeurIPS along with the weak experimental evidence. This would be a good fit for COLT I think. But the improvements included by the authors are good enough to increase the score by one point to 4.

**Limitations:**

Not address

**Paper Formatting Concerns:**

No major concerns

**Quality:**

2

**Strengths And Weaknesses:**

It's possible that the topic of the paper is simply too far outside my expertise for me to understand it totally, but I found this paper very hard to follow. I really think it needs to be significantly reworked for the NeurIPS audience. The paper is pretty idiosyncratic, in my opinion. It has no "Related Works" section for example, and the core contribution only occurs on the last page and is framed as a Theorem, where it really should be a technique. Some points about this can be found in weaknesses.

## Strengths
This paper introduces a new approach to an established task I statistics and thus is of reasonable significance and usefulness. The derivation of their technique is quite extensive, which really precisely and rigorously grounds their technique. Additionally  the work contains several theoretical results, thereby supporting the technique well mathematically.

## Weaknesses
I really think the style of the paper doesn't fit well with NeurIPS. It has a rather strange style where the paper immediately begins explaining the technique it's presenting from the first page and only finishes on the last page. One would typically expect sections providing background, examples, consequences, etc, but virtually the whole paper is a derivation of the technique. I personally don't think this is a good fit for NeurIPS. That being said, I don't really feel like I grasped the paper well, perhaps it is a highly technical topic that can't really be presented any other way; maybe the other reviewers can shine some light on this. I get the impression that this work may fit better in a more core theoretical statistics venue.

A lot of the mathematical objects are not quite familiar enough that I think they can be introduced without any elaboration or are really beyond what most of the community will understand. Other parts are elaborated on too much.
* The introduction to the influence function should be more extensive. I think a couple of examples one trivial (mean) and one less trivial would be good. Is (1.1) really the only definition? Any funciton that satisfies this property is the influence function?
* Concrete applications of the influence function, fully explained, would also be useful to convey the significance of it.
* Some of the exposition is just way beyond what I can understand reasonably easily, a good example is lines 87-97. It really feels like one would need to be quite familiar with the topic to really get whats going on here
* Section 3.1 is unnecessary in my opinion. Much of the community is very familiar with RKHS and this kind of background is unlikely to bring the less familiar to a working familiarity of RKHSs to understand what is going on.
* For papers like these I think its useful for the main text to be a pretty simplified and for the appendix to contain the full technical underpinnings
* It would be good to have the full core contribution technique explained together without any of the extra exposition, I feel like I'm often losing the tread reading this.
### Small Points
* Line 39: is $\psi$ without the $P$ subscript something different?
* line 34: $L_0^2$ is never defined.
* Line 52: What is pathwise differentiable?
* line 168: is the $\partial$ operator different from the $D$ operator in some significant way?
* Line 224: what is a signature? Is it oveloading the variable $K$? or is it somehow the same thing?
* Stylistically it's good to refer to Figures in the main text, not just have them stand alone with no reference. (This may even be in the NeurIPS style guide)


Again I want to highlight that its possible I'm just too far out of my area to expertise to appreciate whats going on in the paper, I will be interested to see what the other reviewers say.

---

> ### Author Rebuttal · Authors · 2025-07-31
>
> ## Organization of the paper
> We thank Reviewer and acknowledge that our presentation can be
> somewhat more streamlined, including a more thorough introduction of the
> background on the influence function (IF) and discussion of prior work, and we
> will do this for the camera-ready version.
> We provide an expanded introduction and motivation at the end of this rebuttal
> as an example and to help with assessing the novelty and contribution of our
> work.
> ## Level of mathematical detail
> Re Paragraph 2 (P2) and Bullet points 3,5,6 (B3,B5,B6) under Weaknesses.
>
> Regarding the suggestion to simplify the presentation in the main text and defer
> all technical underpinnings to the Appendix, we respectfully disagree with the
> reviewer: it is acknowledged in P2 that our problem requires a high level of mathematical
> detail, and in B3 it is suggested that more rather than less introduction of
> core concepts is required to communicate the results of the paper.
> The goal of our paper is to communicate a mathematically novel approach to
> estimating the IF and the ideas that lead us to this result, rather than a
> demonstration of the application of the results with all details and ideas
> hidden in the Appendix.
> Toward this goal, we chose to present our work in a linear fashion, thoroughly
> introducing the background concepts and baseline solution first in Section 1,
> the main theoretical ideas of our proposal in Section 2, and a practical
> implementation of these ideas in Section 3.
> In fact, Reviewers NjaD and em1K highlighted that the paper is well written and
> successfully communicates our ideas despite the highly technical nature of the
> problem, and Reviewer ExzA acknowledges that the technical part of the paper is
> well readable given a general background in analysis and suggests expanding
> the explanation rather than simplifying it.
> Furthermore, Reviewer ExzA acknowledges that our work is within scope for
> NeurIPS, as can similarly be inferred from Reviewers NjaD and em1K, and Reviewer
> 8gPt acknowledges that our work is of good significance for NeurIPS and suggests
> spending significant effort to clarify the background and novelty of our result.
>
> Given this feedback, for camera-ready version we will do
> best to move more technical details to Appendix and will focus on
> expanding Introduction and discussion of our results while still maintaining
> the overall structure.
> ## Necessity of Section 3.1
> We agree that rkHs setting and notation are familiar to most NeurIPS readers,
> we included this section mainly to fix our notation that is needed extensively
> for the rest of Section 3.
> We will shorten this section according to the recommendation of the Reviewer.
> ## More examples/applications
> Influence functions were discovered in statistics while studying the hardness of
> the estimation of a given functional and can be used to find the efficiency
> lower bound and to construct an estimator that achieves this bound.
> An example from Bickel'88 is
> $\theta_0(P) = E[p(X)]$  with IF
> $\psi(x) = 2p(x) - 2\theta(P)$ and an efficient estimator given by
> $\hat\theta=\int\hat p^2 + 2\sum_{i=1}^n[\hat p(X_i) - \int\hat p^2]/n$.
> Here, the IF is needed to both construct and check the efficiency of the estimator.
> Another example are estimands of causal inference, as Reviewer em1K noted.
> Here, the IF of the target parameter is necessary to construct an efficient
> estimator.
> We will expand our experiments to cover examples of such tasks as described in
> response to Reviewer em1K and include $\theta_0$ example in Section 1.
>
> IF is also necessary to study the robustness of the estimand to the
> perturbations of the model.
> For example, in ecological population dynamics models, researchers assume that
> the propensities for capture/recapture and survival follow a logistic model.
> One can ask if the results, say the estimated population size, based on such a
> model are robust to the logistic assumption.
> This can be assessed by relaxing the logistic functional form baked into the
> population model in a way that reveals the bounds on the estimated population
> seize.
> A concrete data application is to construct robustness bounds for the population
> size of Soay sheep estimated in Catchpole'00.
> ## Small points
> Omitted to save space (ask in discussion)
> ## Expanded introduction
>
> A classical result is that MLE is statistically efficient—attains the
> smallest asymptotic variance among “regular” estimators.
> MLE applies to parametric (finite-dimensional) models, where calculations are a
> simple exercises in calculus:
> the derivative of the log-likelihood with respect to the parameter yields the
> score, its variance is the Fisher information, and the inverse of the
> information is the lower bound on the estimator variance.
> Here, the score (up to normalization) is the IF.
>
> However, parametric models are unrealistic, and extending efficiency theory to
> nonparametric (infinite-dimensional) models took decades to develop, culminating
> in foundational texts [Bic+93] and [Vaa00], and recent literature on DML [Che+22].
>
> In nonparametric models, there is no canonical parametrization to carry out the
> analysis.
> Instead, models and estimands (functionals) vary widely in description and often
> require problem-specific techniques for estimation and inference.
> Functional analysis replaces calculus, and solving for efficiency bounds becomes
> a mathematically rich and challenging task.
>
> The central object in this theory is the IF $\psi_P$, which
> characterizes the local sensitivity of a functional $\theta(P)$ to perturbations
> in the distribution $P$.
> The norm of $\psi_P$ gives the lower bound on the variance of regular
> estimators, and $\psi_P$ is the key ingredient for constructing debiased and
> efficient estimators of $\theta(P)$.
> Deriving $\psi_P$ for a given functional is a fundamental task,
> that can be compared to solving a differential equation:
> each model requires specific methods, and finding new IFs is a significant
> contribution.
> Analytic expressions of the IF, as in parametric models, may not even exist.
>
> *How does one estimate the IF?*
> In parametric models, this is mechanical:
> differentiate the log-likelihood, compute the information, normalize the
> score, plug in parameter estimates.
> Recent work by R Agrawal et al, NeurIPS'24 has automated it via autodiff and
> probabilistic programming.
>
> In nonparametric models, this is much more complicated.
> One typically begins by deriving the analytic form of the IF
> using pathwise derivatives along a parametric submodel.
> That is, for a path $P_t$ with known score $g$, the derivative $d\theta(P_t)/dt$
> is computed,
> then one hopes to express it as $E[\phi \cdot g]$ to match the
> representation of Riesz' theorem.
> This result of functional analysis says that any linear map,
> specifically $g\mapsto d\theta(P_t)/dt$, must have such representation with some
> function $\phi$.
> The function $\phi$ obtained via this representation matching is a candidate for
> $\psi_P$.
> The form of $\psi_P$ often includes unknown components (e.g.,
> densities, regression functions) that must then be estimated using appropriate
> statistical methods.
>
> *How does this paper relate to prior work?*
> Several recent works—[CLV19], [Hin+22], [Ken24], [JWZ22, M Jordan et al
> NeurIPS'22]—document the challenge of deriving IFs and propose
> methods for automating this task.
> Specifically, the goal is to remove the functional analysis derivations
> described above, replacing them with standard data analysis tasks such as
> density estimation.
> These works focus on estimating pointwise evaluations of $\psi_P(x)$, using
> the von Mises formula (Theorem 1.1 of our paper; it dates to [Mis47] but
> technical details have been iterated on by many authors).
> However, these methods provide only isolated values, not a functional estimator
> of $\psi_P$, and they lack convergence guarantees in function norms needed for
> downstream tasks like DML.
> By contrast, we develop a completely novel representation that treats $\psi_P$
> as an element of a function space and enables consistent estimation in that
> space.
>
> *What are the contributions of this paper?*
> (I) Spectral representation:
> We derive a regularized representation (Theorem 2.3) that generalizes von
> Mises formula (Theorem 1.1) by replacing pointwise perturbations with an
> expansion into a basis of a Hilbert space $H$.
> This yields a formula for all values of $\psi_P$ simultaneously, rather than
> requiring a separate derivative calculation per point $x$.
> Compared to von Mises, which perturbs the measure $P$ sharply near each
> evaluation point, our method perturbs $P$ along smooth basis functions,
> which is both more numerically stable and aligned with functional approximation
> theory and practice.
>
> (II) Estimation via rkHs approximation and kPCA:
> Using our spectral formula, we construct a data-driven estimator of $\psi_P$ by
> taking $H$ to be an rkHs, and estimating the basis functions
> via kernel PCA—a standard and well-studied (in NeurIPS) data analysis task.
> This construction yields a fully data-driven functional estimator of the IF.
> We prove consistency of this estimator (Theorem 3.7), and argue that this
> approach is practical, scalable, and grounded in rkHs theory and numerical
> analysis.
>
> (III)
> This combination of our spectral representation with kernel PCA enables the use
> of mature and rich theoretical and numerical tools.
> For theory, it allows the use of results on operator concentration.
> For computation, it allows the use of scalable kernel methods (low-rank
> approximations, randomized numerical linear algebra, fast kernel matrix
> computation).
>
> While our paper focuses on consistency, it lays the groundwork for
> analysis of convergence rates, Gaussian approximations, scalable
> computation, and applications to downstream inferences.
> These are all directions for follow-up work.
>
> To our knowledge, no prior work provides a data-driven functional estimator
> of the IF or convergence guarantees in rkHs norms,
> nor has it been shown how to leverage modern kernel-based learning and inference
> pipelines to estimate the IF.

---

> > ### Author Response · Authors · 2025-08-09
> > **Small points (omitted earlier due to space constraints)**
> >
> > *Small points*
> >
> > Line 39: $\psi$ refers to the IF $\psi_P$, we drop the subscript to simplify
> > notation and will remark on this in the paper.
> >
> > Line 34: $L^2_0$ is the subspace of the standard Lebesgue space of square
> > integrable functions containing only functions with zero mean, as concisely
> > explained on line 39. We will clarify this in the paper.
> >
> > Line 52: we introduce pathwise differentiability in a simplified way in the
> > outline of the proof to the Theorem, as indicated on line 65.
> > We will alert the reader to this reverse ordering in introducing the background
> > concepts in our revised introduction to the paper.
> >
> > Line 168: both are derivative operators but they operate on very different
> > objects, $\partial$ stand for the infinitesimal change in a number, and $D$
> > stands for a perturbation to $P$ (infinitesimal change in a function/measure).
> >
> > Line 224: signature is the precise term used to denote the integrand of an
> > integral operator, so we use the same $K$ in two different contexts, as a PSD
> > function and to define an operator, and calling it a kernel in both contexts may
> > lead to confusion.
> >
> > We will refer to our plots in the expanded introduction and the added example of
> > influence functions.

---

> ### Comment · Reviewer_PMrS · 2025-08-05
> **Raise by one point**
>
> Apologies, for the delay, I put this in the mandatory "final justification,” which I didn't know the authors couldn't see.
>
> I appreciate the author's new introduction, which I think is helpful. Nonetheless I find the style, topicality, and presentation somewhat problematic for NeurIPS along with the weak experimental evidence. This would be a good fit for COLT I think. But the improvements included by the authors are good enough to increase the score by one point to 4.

---

> > ### Author Response · Authors · 2025-08-08
> >
> > We want to thank the reviewer for carefully reading the paper, providing
> > insightful feedback and making thoughtful suggestions for improving the
> > exposition which will be incorporated in the revision of the paper.

---

### Note · Authors · 2025-08-13

To re-emphasize the point made in the rebuttal,
we propose a novel solution to the problem studied in [JWZ22, M Jordan,
NeurIPS'22] and [Kandasamy, NeurIPS'15] and contribute to an extensive body of
NeurIPS work that applies IF-based methodology to all domains of ML.
Our results contribute a novel use of the kernel PCA, which is a seminal NeurIPS
technique.

We would like to thank the Reviewers for valuable feedback:
They found:
- our main idea of discretizing the IF in terms of eigenfunctions
improving on the spatial discretization in [JWZ22] is `quite nice`;
- our variational representation that underpins the spectral discretization
nontrivial and original.
- the paper well written overall, reading very well despite being technical in
nature.
- our results to be technically sound and theoretically supported.
- our technical tools well-established and likely to work in practice.

Based on the feedback, we:
- substantially expanded the Introduction to cover the basics of asymptotic
efficiency with examples and explain IF use cases in four paragraphs.
- included a paragraph discussing the analytic derivation of the IF and the
difficulty it poses for practitioners.
- clarified the state of the art and our contribution.
The Introduction should now be accessible, self-contained and informative about
the problem and our contribution for the general NeurIPS audience.

We will:
- expand the explanations of our technical results and move some details
into the Appendix.
- add a discussion of regularization via penalty loading $\lambda$ and
rank $r$ and the computational complexity of our estimator.
- add a How-To section to explain the practical implementation of our
estimator:
computing the eigenfunctions and eigenvalues,
constructing perturbation paths
and computing pathwise derivatives.

In addition to our toy example, we will:
- report experimental results for the standard functionals
in semiparametric efficiency theory:
quantile, average density, information, mutual information, Gini coefficient and
Lorenz value B.
- report experiments that compare the accuracy in downstream tasks of
estimating functionals ATE, RR, OR from page 8 in [Cho+24] using TMLE with the
analytically estimated IF, TMLE with the IF estimated using our method,
and KDPE that does not require the IF.
- report bias-variance trade-offs in our experiments.
- make our plots more readable and explain the interpretations.

---

### Decision · Program_Chairs · 2025-09-17

**Decision:**

Accept (poster)

**Comment:**

The paper proposes an innovative idea to estimate the influence function, and it
theoretically investigates the derived method. The main concern raised in the reviews
was centering around the presentation. Having read the current version, I see that certain
parts can still be improved. On the other hand, the current version's presentation is not
worse than that of some accepted papers I have seen in the past. The experimental section is
short, but as this is mostly a theoretical paper, this is acceptable. Finally, some more theory
like non-compact domains or finite sample bounds would have been nice, but from my
perspective, the presented theory is enough for an innovative paper presenting a new idea.

Despite the issues I have discussed above, the paper is a clear accept for me, as the presented
idea is too nice to be rejected and the presented theory is already sold.